# SOFTMATCH: ADDRESSING THE QUANTITY-QUALITY TRADE-OFF IN SEMI-SUPERVISED LEARNING

**Hao Chen**[1,*] **Ran Tao**[1,*], **Yue Fan**[2], **Yidong Wang**[3]
**Jindong Wang**[3,†] **Bernt Schiele**[2], **Xing Xie**[3], **Bhiksha Raj**[1,4], **Marios Savvides**[1†]

[1]Carnegie Mellon University, [2]Max Planck Institute for Informatics, Saarland Informatics Campus,
[3]Microsoft Research Asia, [4]Mohamed bin Zayed University of AI

## ABSTRACT

The critical challenge of Semi-Supervised Learning (SSL) is how to effectively leverage the limited labeled data and massive unlabeled data to improve the model's generalization performance. In this paper, we first revisit the popular pseudo-labeling methods via a unified sample weighting formulation and demonstrate the inherent quantity-quality trade-off problem of pseudo-labeling with thresholding, which may prohibit learning. To this end, we propose SoftMatch to overcome the trade-off by maintaining both high quantity and high quality of pseudo-labels during training, effectively exploiting the unlabeled data. We derive a truncated Gaussian function to weight samples based on their confidence, which can be viewed as a soft version of the confidence threshold. We further enhance the utilization of weakly-learned classes by proposing a uniform alignment approach. In experiments, SoftMatch shows substantial improvements across a wide variety of benchmarks, including image, text, and imbalanced classification.

## 1 INTRODUCTION

Semi-Supervised Learning (SSL), concerned with learning from a few labeled data and a large amount of unlabeled data, has shown great potential in practical applications for significantly reduced requirements on laborious annotations (Fan et al., 2021; Xie et al., 2020; Sohn et al., 2020; Pham et al., 2021; Zhang et al., 2021; Xu et al., 2021b;a; Chen et al., 2021; Oliver et al., 2018). The main challenge of SSL lies in how to effectively exploit the information of unlabeled data to improve the model's generalization performance (Chapelle et al., 2006). Among the efforts, pseudo-labeling (Lee et al., 2013; Arazo et al., 2020) with confidence thresholding (Xie et al., 2020; Sohn et al., 2020; Xu et al., 2021b; Zhang et al., 2021) is highly-successful and widely-adopted.

The core idea of threshold-based pseudo-labeling (Xie et al., 2020; Sohn et al., 2020; Xu et al., 2021b; Zhang et al., 2021) is to train the model with pseudo-label whose prediction confidence is above a hard threshold, with the others being simply ignored. However, such a mechanism inherently exhibits the *quantity-quality trade-off*, which undermines the learning process. On the one hand, a high confidence threshold as exploited in FixMatch (Sohn et al., 2020) ensures the quality of the pseudo-labels. However, it discards a considerable number of unconfident yet correct pseudo-labels. As an example shown in Fig. 1(a), **around 71% correct pseudo-labels are *excluded* from the training.** On the other hand, dynamically growing threshold (Xu et al., 2021b; Berthelot et al., 2021), or class-wise threshold (Zhang et al., 2021) encourages the utilization of more pseudo-labels but inevitably fully enrolls erroneous pseudo-labels that may mislead training. As an example shown by FlexMatch (Zhang et al., 2021) in Fig. 1(a), **about 16% of the utilized pseudo-labels are *incorrect*.** In summary, the quantity-quality trade-off with a confidence threshold limits the unlabeled data utilization, which may hinder the model's generalization performance.

In this work, we formally define the quantity and quality of pseudo-labels in SSL and summarize the inherent trade-off present in previous methods from a perspective of unified sample weighting for-

---

[*]Equal Contribution: haoc3@andrew.cmu.edu, taoran1@cmu.edu
[†]Correspondence to: jindong.wang@microsoft.com, mariioss@andrew.cmu.edu.

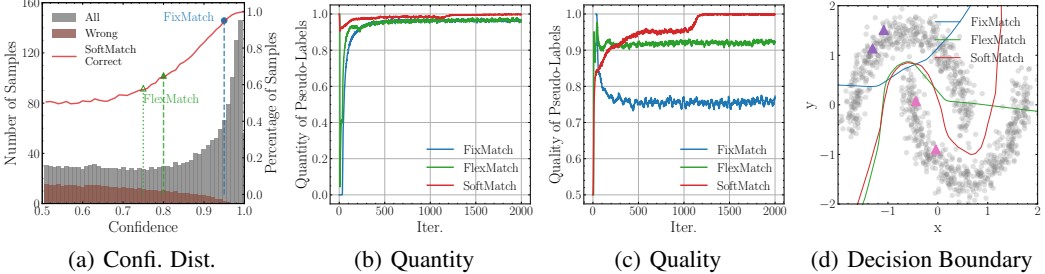

| (a) Confi. Dist. | (b) Quantity | (c) Quality | (d) Decision Boundary |

Figure 1: Illustration on Two-Moon Dataset with only 4 labeled samples (triangle purple/pink points) with others as unlabeled samples in training a 3-layer MLP classifier. Training detail is in Appendix. (a) Confidence distribution, including all predictions and wrong predictions. The red line denotes the correct percentage of samples used by SoftMatch. The part of the line above scatter points denotes the correct percentage for FixMatch (blue) and FlexMatch (green). (b) Quantity of pseudo-labels; (c) Quality of pseudo-labels; (d) Decision boundary. SoftMatch exploits almost all samples during training with lowest error rate and best decision boundary.

mulation. We first identify the fundamental reason behind the quantity-quality trade-off is the lack of sophisticated assumption imposed by the weighting function on the distribution of pseudo-labels. Especially, confidence thresholding can be regarded as a step function assigning binary weights according to samples' confidence, which assumes pseudo-labels with confidence above the threshold are equally correct while others are wrong. Based on the analysis, we propose SoftMatch to overcome the trade-off by maintaining high quantity and high quality of pseudo-labels during training. A truncated Gaussian function is derived from our assumption on the marginal distribution to fit the confidence distribution, which assigns lower weights to possibly correct pseudo-labels according to the deviation of their confidence from the mean of Gaussian. The parameters of the Gaussian function are estimated using the historical predictions from the model during training. Furthermore, we propose Uniform Alignment to resolve the imbalance issue in pseudo-labels, resulting from different learning difficulties of different classes. It further consolidates the quantity of pseudo-labels while maintaining their quality. On the two-moon example, as shown in Fig. 1(c) and Fig. 1(b), SoftMatch achieves a distinctively better accuracy of pseudo-labels while retaining a consistently higher utilization ratio of them during training, therefore, leading to a better-learned decision boundary as shown in Fig. 1(d). We demonstrate that SoftMatch achieves a new state-of-the-art on a wide range of image and text classification tasks. We further validate the robustness of SoftMatch against long-tailed distribution by evaluating imbalanced classification tasks.

Our contributions can be summarized as:

- We demonstrate the importance of the unified weighting function by formally defining the quantity and quality of pseudo-labels, and the trade-off between them. We identify that the inherent trade-off in previous methods mainly stems from the lack of careful design on the distribution of pseudo-labels, which is imposed directly by the weighting function.
- We propose SoftMatch to effectively leverage the unconfident yet correct pseudo-labels, fitting a truncated Gaussian function the distribution of confidence, which overcomes the trade-off. We further propose Uniform Alignment to resolve the imbalance issue of pseudo-labels while maintaining their high quantity and quality.
- We demonstrate that SoftMatch outperforms previous methods on various image and text evaluation settings. We also empirically verify the importance of maintaining the high accuracy of pseudo-labels while pursuing better unlabeled data utilization in SSL.

## 2 REVISIT QUANTITY-QUALITY TRADE-OFF OF SSL

In this section, we formulate the quantity and quality of pseudo-labels from a unified sample weighting perspective, by demonstrating the connection between sample weighting function and the quantity/quality of pseudo-labels. SoftMatch is naturally inspired by revisiting the inherent limitation in quantity-quality trade-off of the existing methods.

## 2.1 PROBLEM STATEMENT

We first formulate the framework of SSL in a $C$-class classification problem. Denote the labeled and unlabeled datasets as $\mathcal{D}_L = \left\{ \mathbf{x}_i^l, \mathbf{y}_i^l \right\}_{i=1}^{N_L}$ and $\mathcal{D}_U = \{ \mathbf{x}_i^u \}_{i=1}^{N_U}$, respectively, where $\mathbf{x}_i^l, \mathbf{x}_i^u \in \mathbb{R}^d$ is the $d$-dimensional labeled and unlabeled training sample, and $\mathbf{y}_i^l$ is the one-hot ground-truth label for labeled data. We use $N_L$ and $N_U$ to represent the number of training samples in $\mathcal{D}_L$ and $\mathcal{D}_U$, respectively. Let $\mathbf{p}(\mathbf{y}|\mathbf{x}) \in \mathbb{R}^C$ denote the model's prediction. During training, given a batch of labeled data and unlabeled data, the model is optimized using a joint objective $\mathcal{L} = \mathcal{L}_s + \mathcal{L}_u$, where $\mathcal{L}_s$ is the supervised objective of the cross-entropy loss ($\mathcal{H}$) on the $B_L$-sized labeled batch:

$$\mathcal{L}_s = \frac{1}{B_L} \sum_{i=1}^{B_L} \mathcal{H}(\mathbf{y}_i, \mathbf{p}(\mathbf{y}|\mathbf{x}_i^l)). \tag{1}$$

For the unsupervised loss, most existing methods with pseudo-labeling (Lee et al., 2013; Arazo et al., 2020; Xie et al., 2020; Sohn et al., 2020; Xu et al., 2021b; Zhang et al., 2021) exploit a confidence thresholding mechanism to mask out the unconfident and possibly incorrect pseudo-labels from training. In this paper, we take a step further and present a *unified* formulation of the confidence thresholding scheme (and other schemes) from the sample weighting perspective. Specifically, we formulate the unsupervised loss $\mathcal{L}_u$ as the **weighted cross-entropy** between the model's prediction of the strongly-augmented data $\Omega(\mathbf{x}^u)$ and pseudo-labels from the weakly-augmented data $\omega(\mathbf{x}^u)$:

$$\mathcal{L}_u = \frac{1}{B_U} \sum_{i=1}^{B_U} \lambda(\mathbf{p}_i)\mathcal{H}(\hat{\mathbf{p}}_i, \mathbf{p}(\mathbf{y}|\Omega(\mathbf{x}_i^u))), \tag{2}$$

where $\mathbf{p}$ is the abbreviation of $\mathbf{p}(\mathbf{y}|\omega(\mathbf{x}^u))$, and $\hat{\mathbf{p}}$ is the one-hot pseudo-label $\mathrm{argmax}(\mathbf{p})$; $\lambda(\mathbf{p})$ is the sample weighting function with range $[0, \lambda_{\max}]$; and $B_U$ is the batch size for unlabeled data.

## 2.2 QUANTITY-QUALITY TRADE-OFF FROM SAMPLE WEIGHTING PERSPECTIVE

In this section, we demonstrate the importance of the unified weighting function $\lambda(\mathbf{p})$, by showing its different instantiations in previous methods and its essential connection with model predictions. We start by formulating the *quantity* and *quality* of pseudo-labels.

**Definition 2.1** (Quantity of pseudo-labels). The quantity $f(\mathbf{p})$ of pseudo-labels enrolled in training is defined as the expectation of the sample weight $\lambda(\mathbf{p})$ over the unlabeled data:

$$f(\mathbf{p}) = \mathbb{E}_{\mathcal{D}_U}[\lambda(\mathbf{p})] \in [0, \lambda_{\max}]. \tag{3}$$

**Definition 2.2** (Quality of pseudo labels). The quality $g(\mathbf{p})$ is the expectation of the weighted 0/1 error of pseudo-labels, assuming the label $\mathbf{y}^u$ is given for $\mathbf{x}^u$ for only theoretical analysis purpose:

$$g(\mathbf{p}) = \sum_i^{N_U} \mathbb{1}(\hat{\mathbf{p}}_i = \mathbf{y}_i^u) \frac{\lambda(\mathbf{p}_i)}{\sum_j^{N_U} \lambda(\mathbf{p}_j)} = \mathbb{E}_{\bar{\lambda}(\mathbf{p})}[\mathbb{1}(\hat{\mathbf{p}} = \mathbf{y}^u)] \in [0, 1], \tag{4}$$

where $\bar{\lambda}(\mathbf{p}) = \lambda(\mathbf{p})/\sum \lambda(\mathbf{p})$ is the probability mass function (PMF) of $\mathbf{p}$ being close to $\mathbf{y}^u$.

Based on the definitions of quality and quantity, we present the *quantity-quality trade-off* of SSL.

**Definition 2.3** (The quantity-quality trade-off). Due to the implicit assumptions of PMF $\bar{\lambda}(\mathbf{p})$ on the marginal distribution of model predictions, the lack of sophisticated design on it usually results in a trade-off in quantity and quality - when one of them increases, the other must decrease. Ideally, a well-defined $\lambda(\mathbf{p})$ should reflect the true distribution and lead to both high quantity and quality.

Despite its importance, $\lambda(\mathbf{p})$ has hardly been defined explicitly or properly in previous methods. In this paper, we first summarize $\lambda(\mathbf{p})$, $\bar{\lambda}(\mathbf{p})$, $f(\mathbf{p})$, and $g(\mathbf{p})$ of relevant methods, as shown in Table 1, with the detailed derivation present in Appendix A.1. For example, naive pseudo-labeling (Lee et al., 2013) and loss weight ramp-up scheme (Samuli & Timo, 2017; Tarvainen & Valpola, 2017; Berthelot et al., 2019b;a) exploit the fixed sample weight to fully enroll all pseudo-labels into training. It is equivalent to set $\lambda = \lambda_{\max}$ and $\bar{\lambda} = 1/N_U$, regardless of $\mathbf{p}$, which means each pseudo-label is assumed equally correct. We can verify the quantity of pseudo-labels is maximized to $\lambda_{\max}$.

Table 1: Summary of different sample weighting function $\lambda(\mathbf{p})$, probability density function $\bar{\lambda}(\mathbf{p})$ of $\mathbf{p}$, quantity $f(\mathbf{p})$ and quality $g(\mathbf{p})$ of pseudo-labels used in previous methods and SoftMatch.

| Scheme | Pseudo-Label | FixMatch | SoftMatch |
|---|---|---|---|
| $\lambda(\mathbf{p})$ | $\lambda_{\max}$ | $\begin{cases} \lambda_{\max}, & \text{if } \max(\mathbf{p}) \geq \tau, \\ 0.0, & \text{otherwise.} \end{cases}$ | $\begin{cases} \lambda_{\max} \exp\left(-\frac{(\max(\mathbf{p})-\mu_t)^2}{2\sigma_t^2}\right), & \text{if } \max(\mathbf{p}) < \mu_t, \\ \lambda_{\max}, & \text{otherwise.} \end{cases}$ |
| $\bar{\lambda}(\mathbf{p})$ | $1/N_U$ | $\begin{cases} 1/\hat{N}_U^\tau, & \text{if } \max(\mathbf{p}) \geq \tau, \\ 0.0, & \text{otherwise.} \end{cases}$ | $\begin{cases} \dfrac{\exp(-\frac{(\max(\mathbf{p}_i)-\hat{\mu}_t)^2}{2\hat{\sigma_t}^2})}{\frac{N_U}{2}+\sum_i^{N_U} \exp(-\frac{(\max(\mathbf{p}_i)-\hat{\mu}_t)^2}{2\hat{\sigma_t}^2})}, & \max(\mathbf{p}) < \mu_t \\ \dfrac{1}{\frac{N_U}{2}+\sum_i^{N_U} \exp(-\frac{(\max(\mathbf{p}_i)-\hat{\mu}_t)^2}{2\hat{\sigma_t}^2})}, & \max(\mathbf{p}) \geq \mu_t \end{cases}$ |
| $f(\mathbf{p})$ | $\lambda_{\max}$ | $\lambda_{\max}\hat{N}_U^\tau/N_U$ | $\lambda_{\max}/2 + \lambda_{\max}/N_U \sum_i^{\frac{N_U}{2}} \exp(-\frac{(\max(\mathbf{p}_i)-\hat{\mu}_t)^2}{2\hat{\sigma_t}^2})$ |
| $g(\mathbf{p})$ | $\sum_i^{N_U} \mathbb{1}(\hat{\mathbf{p}} = \mathbf{y}^u)/N_U$ | $\sum_i^{\hat{N}_U} \mathbb{1}(\hat{\mathbf{p}} = \mathbf{y}^u)/\hat{N}_U^\tau$ | $\sum_j^{\hat{N}_U^{\mu_t}} \mathbb{1}(\hat{\mathbf{p}}_j = \mathbf{y}_j^u)/2\hat{N}_U +$ $\sum_i^{N_U-\hat{N}_U^{\mu_t}} \mathbb{1}(\hat{\mathbf{p}}_i = \mathbf{y}_i^u) \exp(-\frac{(\max(\mathbf{p}_i)-\mu_t)^2}{\sigma_t^2})/2(N_U - \hat{N}_U^{\mu_t})$ |
| Note | High Quantity Low Quality | Low Quantity High Quality | High Quantity High Quality |

However, maximizing quantity also fully involves the erroneous pseudo-labels, resulting in deficient quality, especially in early training. This failure trade-off is due to the implicit uniform assumption on PMF $\bar{\lambda}(\mathbf{p})$ that is far from the realistic situation.

In confidence thresholding (Arazo et al., 2020; Sohn et al., 2020; Xie et al., 2020), we can view the sample weights as being computed from a step function with confidence $\max(\mathbf{p})$ as the input and a pre-defined threshold $\tau$ as the breakpoint. It sets $\lambda(\mathbf{p})$ to $\lambda_{\max}$ when the confidence is above $\tau$ and otherwise 0. Denoting $\hat{N}_U^\tau = \sum_i^{N_U} \mathbb{1}(\max(\mathbf{p}) \geq \tau)$ as the total number of samples whose predicted confidence are above the threshold, $\bar{\lambda}$ is set to a uniform PMF with a total mass of $\hat{N}_U^\tau$ within a fixed range $[\tau, 1]$. This is equal to constrain the unlabeled data as $\hat{\mathcal{D}}_U^\tau = \{\mathbf{x}^u; \max(\mathbf{p}(\mathbf{y}|\mathbf{x}^u)) \geq \tau\}$, with others simply being discarded. We can derive the quantity and the quality as shown in Table 1. A trade-off exists between the quality and quantity of pseudo-labels in confidence thresholding controlled by $\tau$. On the one hand, while a high threshold ensures quality, it limits the quantity of enrolled samples. On the other hand, a low threshold sacrifices quality by fully involving more but possibly erroneous pseudo-labels in training. The trade-off still results from the over-simplification of the PMF from actual cases. Adaptive confidence thresholding (Zhang et al., 2021; Xu et al., 2021b) adopts the dynamic and class-wise threshold, which alleviates the trade-off by evolving the (class-wise) threshold during learning. They impose a further relaxation on the assumption of distribution, but the uniform nature of the assumed PMF remains unchanged.

While some methods indeed consider the definition of $\lambda(\mathbf{p})$ (Ren et al., 2020; Hu et al., 2021; Kim et al., 2022), interestingly, they all neglect the assumption induced on the PMF. The lack of sophisticated modeling of $\bar{\lambda}(\mathbf{p})$ usually leads to a quantity-quality trade-off in the unsupervised loss of SSL, which motivates us to propose SoftMatch to overcome this challenge.

## 3 SOFTMATCH

### 3.1 GAUSSIAN FUNCTION FOR SAMPLE WEIGHTING

Inherently different from previous methods, we generally assume the underlying PMF $\bar{\lambda}(\mathbf{p})$ of marginal distribution follows a *dynamic and truncated Gaussian* distribution of mean $\mu_t$ and variance $\sigma_t$ at $t$-th training iteration. We choose Gaussian for its maximum entropy property and empirically verified better generalization. Note that this is equivalent to treat the deviation of confidence $\max(\mathbf{p})$ from the mean $\mu_t$ of Gaussian as a proxy measure of the correctness of the model's prediction, where samples with higher confidence are less prone to be erroneous than that with lower confidence, consistent to the observation as shown in Fig. 1(a). To this end, we can derive $\lambda(\mathbf{p})$ as:

$$\lambda(\mathbf{p}) = \begin{cases} \lambda_{\max} \exp\left(-\frac{(\max(\mathbf{p})-\mu_t)^2}{2\sigma_t^2}\right), & \text{if } \max(\mathbf{p}) < \mu_t, \\ \lambda_{\max}, & \text{otherwise.} \end{cases} \tag{5}$$

which is also a truncated Gaussian function within the range $[0, \lambda_{\max}]$, on the confidence $\max(\mathbf{p})$.

However, the underlying true Gaussian parameters $\mu_t$ and $\sigma_t$ are still unknown. Although we can set the parameters to fixed values as in FixMatch (Sohn et al., 2020) or linearly interpolate them within some pre-defined range as in Ramp-up (Tarvainen & Valpola, 2017), this might again over-simplify the PMF assumption as discussed before. Recall that the PMF $\bar{\lambda}(\mathbf{p})$ is defined over $\max(\mathbf{p})$, we can instead *fit* the truncated Gaussian function directly to the confidence distribution for better generalization. Specifically, we can estimate $\mu$ and $\sigma^2$ from the historical predictions of the model. At $t$-th iteration, we compute the empirical mean and the variance as:

$$\hat{\mu}_b = \hat{\mathbb{E}}_{B_U}[\max(\mathbf{p})] = \frac{1}{B_U} \sum_{i=1}^{B_U} \max(\mathbf{p}_i),$$

$$\hat{\sigma}_b^2 = \hat{\mathrm{Var}}_{B_U}[\max(\mathbf{p})] = \frac{1}{B_U} \sum_{i=1}^{B_U} (\max(\mathbf{p}_i) - \hat{\mu}_b)^2. \tag{6}$$

We then aggregate the batch statistics for a more stable estimation, using Exponential Moving Average (EMA) with a momentum $m$ over previous batches:

$$\hat{\mu}_t = m\hat{\mu}_{t-1} + (1-m)\hat{\mu}_b,$$

$$\hat{\sigma}_t^2 = m\hat{\sigma}_{t-1}^2 + (1-m)\frac{B_U}{B_U - 1}\hat{\sigma}_b^2, \tag{7}$$

where we use unbiased variance for EMA and initialize $\hat{\mu}_0$ as $\frac{1}{C}$ and $\hat{\sigma}_0^2$ as 1.0. The estimated mean $\hat{\mu}_t$ and variance $\hat{\sigma}_t^2$ are plugged back into Eq. (5) to compute sample weights.

Estimating the Gaussian parameters adaptively from the confidence distribution during training not only improves the generalization but also better resolves the quantity-quality trade-off. We can verify this by computing the quantity and quality of pseudo-labels as shown in Table 1. The derived quantity $f(\mathbf{p})$ is bounded by $[\frac{\lambda_{\max}}{2}(1 + \exp(-\frac{(\frac{1}{C} - \hat{\mu}_t)^2}{2\hat{\sigma}_t^2})), \lambda_{\max}]$, indicating SoftMatch guarantees at least $\lambda_{\max}/2$ of quantity during training. As the model learns better and becomes more confident, i.e., $\hat{\mu}_t$ increases and $\hat{\sigma}_t$ decreases, the lower tail of the quantity becomes much tighter. While quantity maintains high, the quality of pseudo-labels also improves. As the tail of the Gaussian exponentially grows tighter during training, the erroneous pseudo-labels where the model is highly unconfident are assigned with lower weights, and those whose confidence are around $\hat{\mu}_t$ are more efficiently utilized. The truncated Gaussian weighting function generally behaves as **a soft and adaptive version of confidence thresholding**, thus we term the proposed method as SoftMatch.

### 3.2 UNIFORM ALIGNMENT FOR FAIR QUANTITY

As different classes exhibit different learning difficulties, generated pseudo-labels can have potentially *imbalanced* distribution, which may limit the generalization of the PMF assumption (Oliver et al., 2018; Zhang et al., 2021). To overcome this problem, we propose Uniform Alignment (UA), encouraging more uniform pseudo-labels of different classes. Specifically, we define the distribution in pseudo-labels as the expectation of the model predictions on unlabeled data: $\mathbb{E}_{\mathcal{D}_U}[\mathbf{p}(\mathbf{y}|\mathbf{x}^u)]$. During training, it is estimated as $\hat{\mathbb{E}}_{B_U}[\mathbf{p}(\mathbf{y}|\mathbf{x}^u)]$ using the EMA of batch predictions on unlabeled data. We use the ratio between a uniform distribution $\mathbf{u}(C) \in \mathbb{R}^C$ and $\hat{\mathbb{E}}_{B_U}[\mathbf{p}(\mathbf{y}|\mathbf{x}^u)]$ to normalize the each prediction $\mathbf{p}$ on unlabeled data and use the normalized probability to calculate the per-sample loss weight. We formulate the UA operation as:

$$\mathrm{UA}(\mathbf{p}) = \mathrm{Normalize}\left(\mathbf{p} \cdot \frac{\mathbf{u}(C)}{\hat{\mathbb{E}}_{B_U}[\mathbf{p}]}\right), \tag{8}$$

where the $\mathrm{Normalize}(\cdot) = (\cdot)/\sum(\cdot)$, ensuring the normalized probability sums to 1.0. With UA plugged in, the final sample weighting function in SoftMatch becomes:

$$\lambda(\mathbf{p}) = \begin{cases} \lambda_{\max} \exp\left(-\frac{(\max(\mathrm{UA}(\mathbf{p})) - \hat{\mu}_t)^2}{2\hat{\sigma}_t^2}\right), & \text{if } \max(\mathrm{UA}(\mathbf{p})) < \hat{\mu}_t, \\ \lambda_{\max}, & \text{otherwise.} \end{cases} \tag{9}$$

When computing the sample weights, UA encourages larger weights to be assigned to less-predicted pseudo-labels and smaller weights to more-predicted pseudo-labels, alleviating the imbalance issue.

Table 2: Top-1 error rate (%) on CIFAR-10, CIFAR-100, STL-10, and SVHN of 3 different random seeds. Numbers with ∗ are taken from the original papers. The best number is in bold.

| Dataset | CIFAR-10 | | | CIFAR-100 | | | SVHN | | STL-10 | |
|---|---|---|---|---|---|---|---|---|---|---|
| # Label | 40 | 250 | 4,000 | 400 | 2,500 | 10,000 | 40 | 1,000 | 40 | 1,000 |
| PseudoLabel | 74.61$_{\pm0.26}$ | 46.49$_{\pm2.20}$ | 15.08$_{\pm0.19}$ | 87.45$_{\pm0.85}$ | 57.74$_{\pm0.28}$ | 36.55$_{\pm0.24}$ | 64.61$_{\pm5.60}$ | 9.40$_{\pm0.32}$ | 74.68$_{\pm0.99}$ | 32.64$_{\pm0.71}$ |
| MeanTeacher | 70.09$_{\pm1.60}$ | 37.46$_{\pm3.30}$ | 8.10$_{\pm0.21}$ | 81.11$_{\pm1.44}$ | 45.17$_{\pm1.06}$ | 31.75$_{\pm0.23}$ | 36.09$_{\pm3.98}$ | 3.27$_{\pm0.05}$ | 71.72$_{\pm1.45}$ | 33.90$_{\pm1.37}$ |
| MixMatch | 36.19$_{\pm6.48}$ | 13.63$_{\pm0.59}$ | 6.66$_{\pm0.26}$ | 67.59$_{\pm0.66}$ | 39.76$_{\pm0.48}$ | 27.78$_{\pm0.29}$ | 30.60$_{\pm8.39}$ | 3.69$_{\pm0.37}$ | 54.93$_{\pm0.96}$ | 21.70$_{\pm0.68}$ |
| ReMixMatch | 9.88$_{\pm1.03}$ | 6.30$_{\pm0.05}$ | 4.84$_{\pm0.01}$ | 42.75$_{\pm1.05}$ | **26.03$_{\pm0.35}$** | **20.02$_{\pm0.27}$** | 24.04$_{\pm9.13}$ | 5.16$_{\pm0.31}$ | 32.12$_{\pm6.24}$ | 6.74$_{\pm0.14}$ |
| UDA | 10.62$_{\pm3.75}$ | 5.16$_{\pm0.06}$ | 4.29$_{\pm0.07}$ | 46.39$_{\pm1.59}$ | 27.73$_{\pm0.21}$ | 22.49$_{\pm0.23}$ | 5.12$_{\pm4.27}$ | 1.89$_{\pm0.01}$ | 37.42$_{\pm8.44}$ | 6.64$_{\pm0.17}$ |
| FixMatch | 7.47$_{\pm0.28}$ | 4.86$_{\pm0.05}$ | 4.21$_{\pm0.08}$ | 46.42$_{\pm0.82}$ | 28.03$_{\pm0.16}$ | 22.20$_{\pm0.12}$ | 3.81$_{\pm1.18}$ | 1.96$_{\pm0.03}$ | 35.97$_{\pm4.14}$ | 6.25$_{\pm0.33}$ |
| Influence | - | 5.05$_{\pm0.12^*}$ | 4.35$_{\pm0.06^*}$ | - | - | - | 2.63$_{\pm0.23^*}$ | 2.34$_{\pm0.15^*}$ | - | - |
| FlexMatch | 4.97$_{\pm0.06}$ | 4.98$_{\pm0.09}$ | 4.19$_{\pm0.01}$ | 39.94$_{\pm1.62}$ | 26.49$_{\pm0.20}$ | 21.90$_{\pm0.15}$ | 8.19$_{\pm3.20}$ | 6.72$_{\pm0.30}$ | 29.15$_{\pm4.16}$ | 5.77$_{\pm0.18}$ |
| SoftMatch | **4.91$_{\pm0.12}$** | **4.82$_{\pm0.09}$** | **4.04$_{\pm0.02}$** | **37.10$_{\pm0.77}$** | 26.66$_{\pm0.25}$ | 22.03$_{\pm0.03}$ | **2.33$_{\pm0.25}$** | 2.01$_{\pm0.01}$ | **21.42$_{\pm3.48}$** | **5.73$_{\pm0.24}$** |

An essential difference between UA and Distribution Alignment (DA) (Berthelot et al., 2019a) proposed earlier lies in the computation of unsupervised loss. The normalization operation makes the predicted probability biased towards the less-predicted classes. In DA, this might not be an issue, as the normalized prediction is used as *soft target* in the cross-entropy loss. However, with pseudo-labeling, more erroneous pseudo-labels are probably created after normalization, which damages the quality. UA avoids this issue by exploiting original predictions to compute pseudo-labels and normalized predictions to compute sample weights, maintaining both the quantity and quality of pseudo-labels in SoftMatch. The complete training algorithm is shown in Appendix A.2.

## 4 EXPERIMENTS

While most SSL literature performs evaluation on image tasks, we extensively evaluate SoftMatch on various datasets including image and text datasets with classic and long-tailed settings. Moreover, We provide ablation study and qualitative comparison to analyze the effectiveness of SoftMatch. [1]

### 4.1 CLASSIC IMAGE CLASSIFICATION

**Setup**. For the classic image classification setting, we evaluate on CIFAR-10/100 (Krizhevsky et al., 2009), SVHN(Netzer et al., 2011), STL-10 (Coates et al., 2011) and ImageNet (Deng et al., 2009), with various numbers of labeled data, where class distribution of the labeled data is balanced. We use the WRN-28-2 (Zagoruyko & Komodakis, 2016) for CIFAR-10 and SVHN, WRN-28-8 for CIFAR-100, WRN-37-2 (Zhou et al., 2020) for STL-10, and ResNet-50 (He et al., 2016) for ImageNet. For all experiments, we use SGD optimizer with a momentum of $0.9$, where the initial learning rate $\eta_0$ is set to $0.03$. We adopt the cosine learning rate annealing scheme to adjust the learning rate with a total training step of $2^{20}$. The labeled batch size $B_L$ is set to $64$ and the unlabeled batch size $B_U$ is set to 7 times of $B_L$ for all datasets. We set $m$ to $0.999$ and divide the estimated variance $\hat{\sigma}_t$ by 4 for $2\sigma$ of the Gaussian function. We record the EMA of model parameters for evaluation with a momentum of $0.999$. Each experiment is run with three random seeds on labeled data, where we report the top-1 error rate. More details on the hyper-parameters are shown in Appendix A.3.1.

**Results**. SoftMatch obtains the state-of-the-art results on almost all settings in Table 2 and Table 3, except CIFAR-100 with 2,500 and 10,000 labels and SVHN with 1,000 labels, where the results of SoftMatch are comparable to previous methods. Notably, FlexMatch exhibits a performance drop compared to FixMatch on SVHN, since it enrolls too many erroneous pseudo-labels at the beginning of the training that prohibits learning afterward. In contrast, SoftMatch surpasses FixMatch by 1.48% on SVHN with 40 labels, demonstrating its superiority for better utilization of the pseudo-labels. On more realistic datasets, CIFAR-100 with 400 labels, STL-10 with 40 labels, and ImageNet with 10% labels, SoftMatch exceeds FlexMatch by a margin of 7.73%, 2.84%, and 1.33%, respectively. SoftMatch shows the comparable results to FlexMatch on CIFAR-100 with 2,500 and 10,000 labels, whereas ReMixMatch (Berthelot et al., 2019a) demonstrates the best results due to the Mixup (Zhang et al., 2017) and Rotation loss.

---

[1] All experiments in Section 4.1, Section 4.2, and Section 4.5 are conducted with TorchSSL (Zhang et al., 2021) and Section 4.3 are conducted with USB (Wang et al., 2022b) since it only supports NLP tasks back then. More recent results of SoftMatch are included in USB along its updates, refer `https://github.com/Hhhhhhao/SoftMatch` for details.

Table 3: Top1 error rate (%) on ImageNet. The best number is in bold.

| # Label | 100k | 400k |
|---|---|---|
| FixMatch | 43.66 | 32.28 |
| FlexMatch | 41.85 | 31.31 |
| SoftMatch | **40.52** | **29.49** |

Table 4: Top1 error rate (%) on CIFAR-10-LT and CIFAR-100-LT of 5 different random seeds. The best number is in bold.

| Dataset | CIFAR-10-LT | | | CIFAR-100-LT | | |
|---|---|---|---|---|---|---|
| Imbalance $\gamma$ | 50 | 100 | 150 | 20 | 50 | 100 |
| FixMatch | $18.46_{\pm0.30}$ | $25.11_{\pm1.20}$ | $29.62_{\pm0.88}$ | $50.42_{\pm0.78}$ | $57.89_{\pm0.33}$ | $62.40_{\pm0.48}$ |
| FlexMatch | $18.13_{\pm0.19}$ | $25.51_{\pm0.92}$ | $29.80_{\pm0.36}$ | $49.11_{\pm0.60}$ | $57.20_{\pm0.39}$ | $62.70_{\pm0.47}$ |
| SoftMatch | $\mathbf{16.55_{\pm0.29}}$ | $\mathbf{22.93_{\pm0.37}}$ | $\mathbf{27.40_{\pm0.46}}$ | $\mathbf{48.09_{\pm0.55}}$ | $\mathbf{56.24_{\pm0.51}}$ | $\mathbf{61.08_{\pm0.81}}$ |

Table 5: Top1 error rate (%) on text datasets of 3 different random seeds. Best numbers are in bold.

| Datasets | AG News | | DBpedia | | IMDb | Amazon-5 | Yelp-5 |
|---|---|---|---|---|---|---|---|
| # Labels | 40 | 200 | 70 | 280 | 100 | 1000 | 1000 |
| UDA | $16.83_{\pm1.68}$ | $14.34_{\pm1.9}$ | $4.11_{\pm1.44}$ | $6.93_{\pm3.85}$ | $18.33_{\pm0.61}$ | $50.29_{\pm4.6}$ | $47.49_{\pm6.83}$ |
| FixMatch | $17.10_{\pm3.13}$ | $11.24_{\pm1.43}$ | $2.18_{\pm0.92}$ | $1.42_{\pm0.18}$ | $7.59_{\pm0.28}$ | $42.70_{\pm0.53}$ | $39.56_{\pm0.7}$ |
| FlexMatch | $15.49_{\pm1.97}$ | $10.95_{\pm0.56}$ | $2.69_{\pm0.34}$ | $1.69_{\pm0.02}$ | $7.80_{\pm0.23}$ | $42.34_{\pm0.62}$ | $\mathbf{39.01_{\pm0.17}}$ |
| SoftMatch | $\mathbf{12.68_{\pm0.23}}$ | $\mathbf{10.41_{\pm0.13}}$ | $\mathbf{1.68_{\pm0.34}}$ | $\mathbf{1.27_{\pm0.1}}$ | $\mathbf{7.48_{\pm0.12}}$ | $42.14_{\pm0.92}$ | $39.31_{\pm0.45}$ |

## 4.2 LONG-TAILED IMAGE CLASSIFICATION

**Setup**. We evaluate SoftMatch on a more realistic and challenging setting of imbalanced SSL (Kim et al., 2020; Wei et al., 2021; Lee et al., 2021; Fan et al., 2022), where both the labeled and the unlabeled data exhibit long-tailed distributions. Following (Fan et al., 2022), the imbalance ratio $\gamma$ ranges from 50 to 150 and 20 to 100 for CIFAR-10-LT and CIFAR-100-LT, respectively. Here, $\gamma$ is used to exponentially decrease the number of samples from class 0 to class $C$ (Fan et al., 2022). We compare SoftMatch with two strong baselines: FixMatch (Sohn et al., 2020) and FlexMatch (Zhang et al., 2021). All experiments use the same WRN-28-2 (Zagoruyko & Komodakis, 2016) as the backbone and the same set of common hyper-parameters. Each experiment is repeated five times with different data splits, and we report the average test accuracy and the standard deviation. More details are in Appendix A.3.2.

**Results**. As is shown in Table 4, SoftMatch achieves the best test error rate across all long-tailed settings. The performance improvement over the previous state-of-the-art is still significant even at large imbalance ratios. For example, SoftMatch outperforms the second-best by 2.4% at $\gamma = 150$ on CIFAR-10-LT, which suggests the superior robustness of our method against data imbalance.

**Discussion**. Here we study the design choice of uniform alignment as it plays a key role in Soft-Match's performance on imbalanced SSL. We conduct experiments with different target distributions for alignment. Specifically, the default uniform target distribution $\mathbf{u}(C)$ can be replaced by ground-truth class distribution or the empirical class distribution estimated by seen labeled data during training. The results in Fig. 3(a) show a clear advantage of using uniform distribution. Uniform target distribution enforces the class marginal to become uniform, which has a strong regularization effect of balancing the head and tail classes in imbalanced classification settings.

## 4.3 TEXT CLASSIFICATION

**Setup**. In addition to image classification tasks, we further evaluate SoftMatch on text topic classification tasks of AG News and DBpedia, and sentiment tasks of IMDb, Amazon-5, and Yelp-5 (Maas et al., 2011; Zhang et al., 2015). We split a validation set from the training data to evaluate the algorithms. For Amazon-5 and Yelp-5, we randomly sample 50,000 samples per class from the training data to reduce the training time. We fine-tune the pre-trained BERT-Base (Devlin et al., 2018) model for all datasets using UDA (Xie et al., 2020), FixMatch (Sohn et al., 2020), FlexMatch (Zhang et al., 2021), and SoftMatch. We use AdamW (Kingma & Ba, 2014; Loshchilov & Hutter, 2017) optimizer with an initial learning rate of $1e-5$ and the same cosine scheduler as image classification tasks. All algorithms are trained for a total iteration of $2^{18}$. The fine-tuned model is directly used for evaluation rather than the EMA version. To reduce the GPU memory usage, we set both $B_L$ and $B_U$ to 16. Other algorithmic hyper-parameters stay the same as image classification tasks. Details of the data splitting and the hyper-parameter used are in Appendix A.3.3.

**Results**. The results on text datasets are shown in Table 5. SoftMatch consistently outperforms other methods, especially on the topic classifications tasks. For instance, SoftMatch achieves an error rate

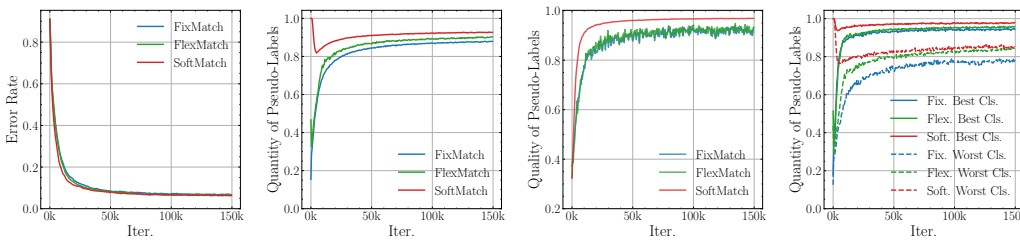

Figure 2: Qualitative analysis of FixMatch, FlexMatch, and SoftMatch on CIFAR-10 with 250 labels. (a) Evaluation error; (b) Quantity of Pseudo-Labels; (c) Quality of Pseudo-Labels; (d) Quality of Pseudo-Labels from the best and worst learned class. Quality is computed according to the underlying ground truth labels. SoftMatch achieves significantly better performance.

of $12.68\%$ on AG news with only 40 labels and $1.68\%$ on DBpedia with 70 labels, surpassing the second best by a margin of $2.81\%$ and $0.5\%$ respectively. On sentiment tasks, SoftMatch also shows the best results on Amazon-5 and IMDb, and comparable results to its counterpart on Yelp-5.

## 4.4 QUALITATIVE ANALYSIS

In this section, we provide a qualitative comparison on CIFAR-10 with 250 labels of FixMatch (Sohn et al., 2020), FlexMatch (Zhang et al., 2021), and SoftMatch from different aspects, as shown in Fig. 2. We compute the error rate, the quantity, and the quality of pseudo-labels to analyze the proposed method, using the ground truth of unlabeled data that is unseen during training.

**SoftMatch utilizes the unlabeled data better**. From Fig. 2(b) and Fig. 2(c), one can observe that SoftMatch obtains highest quantity and quality of pseudo-labels across the training. Larger error with more fluctuation is present in quality of FixMatch and FlexMatch due to the nature of confidence thresholding, where significantly more wrong pseudo-labels are enrolled into training, leading to larger variance in quality and thus unstable training. While attaining a high quality, SoftMatch also substantially improves the unlabeled data utilization ratio, i.e., the quantity, as shown in Fig. 2(b), demonstrating the design of truncated Gaussian function could address the quantity-quality trade-off of the pseudo-labels. We also present the quality of the best and worst learned classes, as shown in Fig. 2(d), where both retain the highest along training in SoftMatch. The well-solved quantity-quality trade-off allows **SoftMatch achieves better performance on convergence and error rate**, especially for the first 50k iterations, as in Fig. 2(a).

## 4.5 ABLATION STUDY

**Sample Weighting Functions**. We validate different instantiations of $\lambda(\mathbf{p})$ to verify the effectiveness of the truncated Gaussian assumption on PMF $\lambda(\bar{\mathbf{p}})$, as shown in Fig. 3(b). Both linear function and Quadratic function fail to generalize and present large performance gap between Gaussian due to the naive assumption on PMF as discussed before. Truncated Laplacian assumption also works well on different settings, but truncated Gaussian demonstrates the most robust performance.

**Gaussian Parameter Estimation**. SoftMatch estimates the Gaussian parameters $\mu$ and $\sigma^2$ directly from the confidence generated from all unlabeled data along the training. Here we compare it (*All-Class*) with two alternatives: (1) *Fixed*: which uses pre-defined $\mu$ and $\sigma^2$ of 0.95 and 0.01. (2) *Per-Class*: where a Gaussian for each class instead of a global Gaussian weighting function. As shown in Fig. 3(c), the inferior performance of *Fixed* justifies the importance of adaptive weight adjustment in SoftMatch. Moreover, *Per-Class* achieves comparable performance with SoftMatch at 250 labels, but significantly higher error rate at 40 labels. This is because an accurate parameter estimation requires many predictions for each class, which is not available for *Per-Class*.

**Uniform Alignment on Gaussian**. To verify the impact of UA, we compare the performance of SoftMatch with and without UA, denoted as all-class with UA and all-class without UA in Fig. 3(d). Since the per-class estimation standalone can also be viewed as a way to achieve fair class utilization (Zhang et al., 2021), we also include it in comparison. Removing UA from SoftMatch has a slight performance drop. Besides, per-class estimation produces significantly inferior results on SVHN.

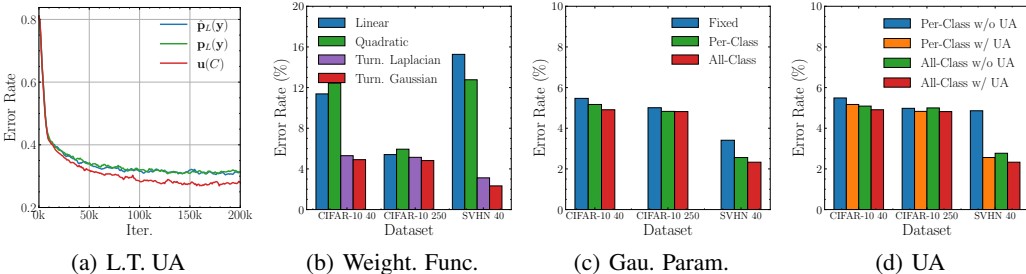

| (a) L.T. UA | (b) Weight. Func. | (c) Gau. Param. | (d) UA |

Figure 3: Ablation study of SoftMatch. (a) Target distributions for Uniform Alignment (UA) on long-tailed setting; (b) Error rate of different sample functions; (c) Error rate of different Gaussian parameter estimation, with UA enabled; (d) Ablation on UA with Gaussian parameter estimation;

We further include the detailed ablation of sample functions and several additional ablation study in Appendix A.5 due to space limit. These studies demonstrate that SoftMatch stays robust to different EMA momentum, variance range, and UA target distributions on balanced distribution settings.

## 5 RELATED WORK

Pseudo-labeling (Lee et al., 2013) generates artificial labels for unlabeled data and trains the model in a self-training manner. Consistency regularization (Samuli & Timo, 2017) is proposed to achieve the goal of producing consistent predictions for similar data points. A variety of works focus on improving the pseudo-labeling and consistency regularization from different aspects, such as loss weighting (Samuli & Timo, 2017; Tarvainen & Valpola, 2017; Iscen et al., 2019; Ren et al., 2020), data augmentation (Grandvalet et al., 2005; Sajjadi et al., 2016; Miyato et al., 2018; Berthelot et al., 2019b;a; Xie et al., 2020; Cubuk et al., 2020; Sajjadi et al., 2016), label allocation (Tai et al., 2021), feature consistency (Li et al., 2021; Zheng et al., 2022; Fan et al., 2021), and confidence thresholding (Sohn et al., 2020; Zhang et al., 2021; Xu et al., 2021b).

Loss weight ramp-up strategy is proposed to balance the learning on labeled and unlabeled data. (Samuli & Timo, 2017; Tarvainen & Valpola, 2017; Berthelot et al., 2019b;a). By progressively increasing the loss weight for the unlabeled data, which prevents the model involving too much ambiguous unlabeled data at the early stage of training, the model therefore learns in a curriculum fashion. Per-sample loss weight is utilized to better exploit the unlabeled data (Iscen et al., 2019; Ren et al., 2020). The previous work "Influence" shares a similar goal with us, which aims to calculate the loss weight for each sample but for the motivation that not all unlabeled data are equal (Ren et al., 2020). SAW (Lai et al., 2022) utilizes effective weights (Cui et al., 2019) to overcome the class-imbalanced issues in SSL. Modeling of loss weight has also been explored in semi-supervised segmentation (Hu et al., 2021). De-biased self-training (Chen et al., 2022; Wang et al., 2022a) study the data bias and training bias brought by involving pseudo-labels into training, which is similar exploration of quantity and quality in SoftMatch. Kim et al. (2022) proposed to use a small network to predict the loss weight, which is orthogonal to our work.

Confidence thresholding methods (Sohn et al., 2020; Xie et al., 2020; Zhang et al., 2021; Xu et al., 2021b) adopt a threshold to enroll the unlabeled samples with high confidence into training. Fix-Match (Sohn et al., 2020) uses a fixed threshold to select pseudo-labels with high quality, which limits the data utilization ratio and leads to imbalanced pseudo-label distribution. Dash (Xu et al., 2021b) gradually increases the threshold during training to improve the utilization of unlabeled data. FlexMatch (Zhang et al., 2021) designs class-wise thresholds and lowers the thresholds for classes that are more difficult to learn, which alleviates class imbalance.

## 6 CONCLUSION

In this paper, we revisit the quantity-quality trade-off of pseudo-labeling and identify the core reason behind this trade-off from a unified sample weighting. We propose SoftMatch with truncated Gaussian weighting function and Uniform Alignment that overcomes the trade-off, yielding both high quantity and quality of pseudo-labels during training. Extensive experiments demonstrate the effectiveness of our method on various tasks. We hope more works can be inspired in this direction, such as designing better weighting functions that can discriminate correct pseudo-labels better.

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

# A APPENDIX

## A.1 QUANTITY-QUALITY TRADE-OFF

In this section, we present the detailed definition and derivation of the quantity and quality formulation. Importantly, we identify that the sampling weighting function $\lambda(\mathbf{p}) \in [0, \lambda_{\max}]$ is directly related to the (implicit) assumption of probability mass function (PMF) over $\mathbf{p}$ for $\mathbf{p} \in \{\mathbf{p}(\mathbf{y}|\mathbf{x}^u); \mathbf{x}^u \in \mathcal{D}_U\}$, i.e., the distribution of $\mathbf{p}$. From the unified sample weighting function perspective, we show the analysis of quantity and quality of the related methods and SoftMatch.

### A.1.1 QUANTITY AND QUALITY

**Derivation Definition 2.1**

The definition and derivation of quantity $f(\mathbf{p})$ of pseudo-labels is rather straightforward. We define the quantity as the percentage/ratio of unlabeled data enrolled in the weighted unsupervised loss. In other words, the quantity is the average sample weights on unlabeled data:

$$f(\mathbf{p}) = \sum_i^{N_U} \frac{\lambda(\mathbf{p}_i)}{N_U} = \mathbb{E}_{\mathcal{D}_U}[\lambda(\mathbf{p}_i)], \tag{10}$$

where each unlabeled data is uniformly sampled from $\mathcal{D}_U$ and $f(\mathbf{p}) \in [0, \lambda_{\max}]$.

**Derivation Definition 2.2**

We define the quality $g(\mathbf{p})$ of pseudo-labels as the percentage/ratio of correct pseudo-labels enrolled in the weighted unsupervised loss, assuming the ground truth label $\mathbf{y}^u$ of unlabeled data is known. With the 0/1 correct indicator function $\gamma(\mathbf{p})$ being defined as:

$$\gamma(\mathbf{p}) = \mathbb{1}(\hat{\mathbf{p}} = \mathbf{y}^u) \in \{0, 1\}, \tag{11}$$

where $\hat{\mathbf{p}}$ is the one-hot vector of pseudo-label $\mathrm{argmax}(\mathbf{p})$. We can formulate quality as:

$$\begin{aligned} g(\mathbf{p}) &= \sum_i^{N_U} \gamma(\mathbf{p}_i) \frac{\lambda(\mathbf{p}_i)}{\sum_j^{N_U} \lambda(\mathbf{p}_j)} \\ &= \sum_i^{N_U} \gamma(\mathbf{p}_i) \bar{\lambda}(\mathbf{p}_i) \\ &= \mathbb{E}_{\bar{\lambda}(\mathbf{p})}[\gamma(\mathbf{p})] \\ &= \mathbb{E}_{\bar{\lambda}(\mathbf{p})}[\mathbb{1}(\hat{\mathbf{p}} = \mathbf{y}^u)] \in [0, 1]. \end{aligned} \tag{12}$$

We denote $\bar{\lambda}(\mathbf{p})$ as the probability mass function (PMF) of $\mathbf{p}$, with $\bar{\lambda}(\mathbf{p}) \geq 0$ and $\sum \bar{\lambda}(\mathbf{p}) = 1.0$.

This indicates that, once $\lambda(\mathbf{p})$ is set to a function, the assumption on the PMF of $\mathbf{p}$ is made. In most of the previous methods (Tarvainen & Valpola, 2017; Berthelot et al., 2019b;a; Sohn et al., 2020; Zhang et al., 2021; Xu et al., 2021b), although they do not explicitly set $\lambda(\mathbf{p})$, the introduction of loss weight schemes implicitly relates to the PMF of $\mathbf{p}$. While the ground truth label $\mathbf{p}$ is actually unknown in practice, we can still use it for theoretical analysis.

In the following sections, we explicitly derive the sampling weighting function $\lambda(\mathbf{p})$, probability mass function $\bar{\lambda}(\mathbf{p})$, quantity $f(\mathbf{p})$, and quality $g(\mathbf{p})$ for each relevant method.

### A.1.2 NAIVE PSEUDO-LABELING

In naive pseudo-labeling (Lee et al., 2013), the pseudo-labels are directly used to the model itself. This is equivalent to set $\lambda(\mathbf{p})$ to a fixed value $\lambda_{\max}$, which is a hyper-parameter. We can write:

$$\lambda(\mathbf{p}) = \lambda_{\max}, \tag{13}$$

$$\bar{\lambda}(\mathbf{p}) = \frac{\lambda_{\max}}{N_U \lambda_{\max}} = \frac{1}{N_U}, \tag{14}$$

$$f(\mathbf{p}) = \sum_i^{N_U} \frac{\lambda_{\max}}{N_U} = \lambda_{\max}, \tag{15}$$

$$g(\mathbf{p}) = \sum_i^{N_U} \frac{\mathbb{1}(\hat{\mathbf{p}}_i = \mathbf{y}_i^u)}{N_U}. \tag{16}$$

We can observe that the naive self-training maximizes the quantity of the pseudo-labels by fully enrolling them into training. However, full enrollment results in pseudo-labels of low quality. At beginning of training, a large portion of the pseudo-labels would be wrong, i.e., $\gamma(\mathbf{p}) = 0$, since the model is not well-learned. The wrong pseudo-labels usually leads to confirmation bias (Guo et al., 2017; Arazo et al., 2020) as training progresses, where the model memorizes the wrong pseudo-labels and becomes very confident on them. We can also notice that, by setting $\lambda(\mathbf{p})$ to a fixed value $\lambda_{\max}$, we implicitly assume the PMF of the model's prediction $\mathbf{p}$ is uniform, which is far away from the realistic distribution.

### A.1.3 LOSS WEIGHT RAMP UP

In the earlier attempts of semi-supervised learning, a bunch of work (Tarvainen & Valpola, 2017; Berthelot et al., 2019b;a) exploit the loss weight ramp up technique to avoid involving too much erroneous pseudo-labels in the early training and let the model focus on learning from labeled data first. In this case, the sample weighting function is formulated as a function of training iteration $t$, which is linearly increased during training and reaches its maximum $\lambda_{\max}$ after $T$ warm-up iterations. Thus we have:

$$\lambda(\mathbf{p}) = \lambda_{\max} \min(\frac{t}{T}, 1), \tag{17}$$

$$\bar{\lambda}(\mathbf{p}) = \frac{\lambda_{\max} \min(\frac{t}{T}, 1)}{N_U \lambda_{\max} \min(\frac{t}{T}, 1)} = \frac{1}{N_U}, \tag{18}$$

$$f(\mathbf{p}) = \lambda_{\max} \min(\frac{t}{T}, 1), \tag{19}$$

$$g(\mathbf{p}) = \sum_i^{N_U} \frac{\mathbb{1}(\hat{\mathbf{p}}_i = \mathbf{y}_i^u)}{N_U}, \tag{20}$$

which demonstrates the same uniform assumption of PMF and same quality function as naive self-training. It also indicates that, as long as same sample weight is used for all unlabeled data, a uniform assumption of PDF over $\mathbf{p}$ is made.

### A.1.4 FIXED CONFIDENCE THRESHOLDING

Confidence thresholding introduces a filtering mechanism, where the unlabeled data whose prediction confidence $\max(\mathbf{p})$ is above the pre-defined threshold $\tau$ is fully enrolled during training, and others being ignored (Xie et al., 2020; Sohn et al., 2020). The confidence thresholding mechanism can be formulated by setting $\lambda(\mathbf{p})$ as a step function - when the confidence is above threshold, the

sample weight is set to $\lambda_{\max}$, and otherwise 0. We can derive:

$$\lambda(\mathbf{p}) = \begin{cases} \lambda_{\max}, & \text{if } \max(\mathbf{p}) \geq \tau, \\ 0.0, & \text{otherwise.} \end{cases} \tag{21}$$

$$\bar{\lambda}(\mathbf{p}) = \frac{\mathbb{1}(\max(\mathbf{p}) \geq \tau)}{\sum_i^{N_U} \mathbb{1}(\max(\mathbf{p}_i) \geq \tau)} = \begin{cases} \frac{1}{\hat{N}_U}, & \text{if } \max(\mathbf{p}) \geq \tau, \\ 0.0, & \text{otherwise.} \end{cases} \tag{22}$$

$$f(\mathbf{p}) = \sum_i^{N_U} \frac{\mathbb{1}(\max(\mathbf{p}_i) \geq \tau)\lambda_{\max}}{N_U} = \lambda_{\max} \frac{\hat{N}_U}{N_U}, \tag{23}$$

$$g(\mathbf{p}) = \sum_i^{\hat{N}_U} \frac{\mathbb{1}(\hat{\mathbf{p}}_i = \mathbf{y}_i^u)}{\hat{N}_U}, \tag{24}$$

$$\tag{25}$$

where we set $\hat{N}_U = \sum_i^{N_U} \mathbb{1}(\max(\mathbf{p}_i) \geq \tau)$, i.e., number of unlabeled samples whose prediction confidence $\max(\mathbf{p})$ are above threshold $\tau$.

Interestingly, one can find that confidence thresholding directly modeling the PMF over the prediction confidence $\max(\mathbf{p})$. Although it still makes the uniform assumption, as shown in Eq. (22), it constrains the probability mass to concentrate in the range of $[\tau, 1]$. As the model is more confident about the pseudo-labels, and the unconfident ones are excluded from training, it is more likely that $\hat{\mathbf{p}}$ would be close to $\mathbf{y}^u$, thus ensuring the quality of the pseudo-labels to a high value if a high threshold is exploited. However, a higher threshold corresponds to smaller $\hat{N}_U$, directly reducing the quantity of pseudo-labels. We can clearly observe a trade-off between quantity and quality of using fixed confidence thresholding. In addition, assuming the PMF of $\max(\mathbf{p})$ as a uniform within a range $[\tau, 1]$ still does not reflect the actually distribution over confidence during training.

### A.1.5 SOFTMATCH

In this paper, we propose SoftMatch to overcome the trade-off between quantity and quality of pseudo-labels. Different from previous methods, which implicitly make over-simplified assumptions on the distribution of $\mathbf{p}$, we directly modelling the PMF of $\max(\mathbf{p})$, from which we derive the sample weighting function $\lambda(\mathbf{p})$ used in SoftMatch.

We assume the confidence of model predictions $\max(\mathbf{p})$ generally follows the Gaussian distribution $\mathcal{N}(\max(\mathbf{p}); \hat{\mu}_t, \hat{\sigma}_t)$ when $\max(\mathbf{p}) < \mu_t$ and the uniform distribution when $\max(\mathbf{p}) \geq \mu_t$. Note that $\mu_t$ and $\sigma_t$ is changing along training as the model learns better. One can see that the uniform part of the PMF is similar to that of confidence thresholding, and it is the Gaussian part makes SoftMatch distinct from previous methods. In SoftMatch, we directly estimate the Gaussian parameters on $\max(\mathbf{p})$ using Maximum Likelihood Estimation (MLE), rather than set them to fixed values, which is more consistent to the actual distribution of prediction confidence. Using the definition of PMF $\bar{\lambda}(\mathbf{p})$, we can directly write the sampling weighting function $\lambda(\mathbf{p})$ of SoftMatch as:

$$\lambda(\mathbf{p}) = \begin{cases} \lambda_{\max}\sqrt{2\pi}\sigma_t\phi(\max(\mathbf{p}; \mu_t, \sigma_t)), & \max(\mathbf{p}) < \mu_t \\ \lambda_{\max}, & \max(\mathbf{p}) \geq \mu_t \end{cases}, \tag{26}$$

where $\phi(x; \mu, \sigma) = \frac{1}{\sqrt{2\pi}\sigma} \exp(-\frac{(x-\mu)^2}{2\sigma^2})$. Without loss of generality, we can assume $\max(\mathbf{p}_i) < \mu_t$ for $i \in [0, \frac{N_U}{2}]$, as $\mu_t = \frac{1}{N_U}\sum_i^{N_U} \max(\mathbf{p}_i)$ (shown in Eq. (6)) and thus $\mathcal{P}(\max(\mathbf{p}) < \mu_t) = 0.5$.

Therefore, $\sum \lambda(\mathbf{p})$ is computed as follows:

$$
\begin{aligned}
\sum_{i}^{N_U} \lambda(\mathbf{p}_i) \\
&= \sum_{i=1}^{\frac{N_U}{2}} \lambda(\mathbf{p}_i) + \sum_{j=\frac{N_U}{2}+1}^{N_U} \lambda(\mathbf{p}_j) \\
&= \sum_{i}^{\frac{N_U}{2}} \lambda_{\max} \sqrt{2\pi} \sigma_t \phi(\max(\mathbf{p}_i); \mu_t, \sigma_t)) + \sum_{j=\frac{N_U}{2}+1}^{N_U} \lambda_{\max} \\
&= \lambda_{\max} \left( \frac{N_U}{2} + \sum_{i}^{\frac{N_U}{2}} \exp(-\frac{(\max(\mathbf{p}_i) - \mu_t)^2}{2\sigma_t^2}) \right)
\end{aligned}
\tag{27}
$$

Further,

$$
\begin{aligned}
f(\mathbf{p}) &= \frac{1}{N_U} \sum_{i}^{N_U} \lambda(\mathbf{p}_i) \\
&= \frac{1}{N_U} \left( \sum_{i=1}^{\frac{N_U}{2}} \lambda(\mathbf{p}_i) + \sum_{j=\frac{N_U}{2}+1}^{N_U} \lambda(\mathbf{p}_j) \right) \\
&= \frac{\lambda_{\max}}{N_U} \left( \frac{N_U}{2} + \sum_{j}^{\frac{N_U}{2}} \exp(-\frac{(\max(\mathbf{p}_j) - \mu_t)^2}{2\sigma_t^2}) \right) \\
&= \frac{\lambda_{\max}}{2} + \frac{\lambda_{\max}}{N_U} \sum_{j}^{\frac{N_U}{2}} \exp(-\frac{(\max(\mathbf{p}_j) - \mu_t)^2}{2\sigma_t^2})
\end{aligned}
\tag{28}
$$

Since $\max(\mathbf{p}_i) < \mu_t$ for $i \in [0, \frac{N_U}{2}]$,

$$
\exp(-\frac{(\frac{1}{C} - \mu_t)^2}{2\sigma_t^2}) <= \exp(-\frac{(\max(\mathbf{p}_i) - \mu_t)^2}{2\sigma_t^2}) < 1
$$

$$
\frac{N_U}{2} \exp(-\frac{(\frac{1}{C} - \mu_t)^2}{2\sigma_t^2}) <= \sum_{i}^{\frac{N_U}{2}} \exp(-\frac{(\max(\mathbf{p}_i) - \mu_t)^2}{2\sigma_t^2}) < \frac{N_U}{2}
$$

$$
\frac{\lambda_{\max}}{2} < \frac{\lambda_{\max}}{2}(1 + \exp(-\frac{(\frac{1}{C} - \mu_t)^2}{2\sigma_t^2})) <= f(\mathbf{p}) < \lambda_{\max}
$$

Therefore, SoftMatch can guarantee at least half of the possible contribution to the final loss, improving the utilization of unlabeled data. Besides, as $\sigma_t$ is also estimated from $\max(\mathbf{p})$, the lower bound of $f(\mathbf{p})$ would become tighter during training with a better and more confident model.

With the derived $\sum \lambda(\mathbf{p})$, We can write the PDF $\bar{\lambda}(\mathbf{p})$ in SoftMatch as:

$$
\bar{\lambda}(\mathbf{p}) = \begin{cases} \frac{\sqrt{2\pi}\sigma_t \phi(\max(\mathbf{p}); \mu_t, \sigma_t)}{\frac{N_U}{2} + \sum_i^{\frac{N_U}{2}} \sqrt{2\pi}\sigma_t \phi(\max(\mathbf{p}); \mu_t, \sigma_t)}, & \max(\mathbf{p}) < \mu_t \\ \frac{1}{\frac{N_U}{2} + \sum_i^{\frac{N_U}{2}} \sqrt{2\pi}\sigma_t \phi(\max(\mathbf{p}); \mu_t, \sigma_t)}, & \max(\mathbf{p}) \geq \mu_t \end{cases},
\tag{29}
$$

and further derive the quality of pseudo-labels in SoftMatch as:

$$
\begin{aligned}
g(\mathbf{p}) &= \sum_i^{N_U} \mathbb{1}(\hat{\mathbf{p}}_i = \mathbf{y}^u)\bar{\lambda}(\mathbf{p}) \\
&= \frac{1}{\sum_k^{N_U} \lambda(\mathbf{p}_k)} \sum_i^{N_U} \gamma(\mathbf{p}_i)\lambda(\mathbf{p}_i) \\
&= \frac{1}{\sum_k^{N_U} \lambda(\mathbf{p}_k)} \left( \sum_i^{\frac{N_U}{2}} \gamma(\mathbf{p}_i)\lambda(\mathbf{p}_i) + \sum_{j=\frac{N_U}{2}+1}^{\frac{N_U}{2}} \gamma(\mathbf{p}_j)\lambda(\mathbf{p}_j) \right) \\
&= \sum_i^{\frac{N_U}{2}} \gamma(\mathbf{p}_i)\frac{\lambda_{\max}\sqrt{2\pi}\sigma_t\phi(\max(\mathbf{p}_i);\mu_t,\sigma_t)}{\sum_k^{N_U} \lambda(\mathbf{p}_k)} + \sum_j^{\frac{N_U}{2}} \gamma(\mathbf{p}_j)\frac{\lambda_{\max}}{\sum_k^{N_U} \lambda(\mathbf{p}_k)} \\
&= \sum_i^{N_U - \hat{N}_U} \frac{\mathbb{1}(\hat{\mathbf{p}}_i = \mathbf{y}_i^u)\exp(-\frac{(\max(\mathbf{p}_i)-\mu_t)^2}{\sigma_t^2})}{2(N_U - \hat{N}_U)} + \sum_j^{\hat{N}_U} \frac{\mathbb{1}(\hat{\mathbf{p}}_j = \mathbf{y}_j^u)}{2\hat{N}_U}
\end{aligned}
\tag{30}
$$

where $\hat{N}_U = \sum_i^{N_U} \mathbb{1}(\max(\mathbf{p}_i) \geq \mu_t)$. From the above equation, we can see that for pseudo-labels whose confidence is above $\mu_t$, the quality is as high as in confidence thresholding; for pseudo-labels whose confidence is lower, thus more possible to be erroneous, the quality is weighted by the deviation from $\mu_t$.

At the beginning of training, where the model is unconfident about most of the pseudo-labels, Soft-Match guarantees the quantity for at least $\frac{\lambda_{\max}}{2}$ and high quality for at least $\sum_j^{\hat{N}_U} \frac{\mathbb{1}(\hat{\mathbf{p}}_j = \mathbf{y}_j^u)}{2\hat{N}_U}$. As the model learns better and becomes more confident, i.e., $\mu_t$ increases and $\sigma_t$ decreases, the lower bound of quantity becomes tighter. The increase in $\hat{N}_U$ leads to better quality with pseudo-labels whose confidence below $\mu_t$ are further down-weighted. Therefore, SoftMatch overcomes the quantity-quality trade-off.

## A.2 ALGORITHM

We present the pseudo algorithms of SoftMatch in this section. SoftMatch adopts the truncated Gaussian function with parameters estimated from the EMA of the confidence distribution at each training step, which introduce trivial computations.

---

**Algorithm 1** SoftMatch algorithm.

1: **Input:** Number of classes $C$, labeled batch $\{\mathbf{x}_i, \mathbf{y}_i\}_{i\in[B_L]}$, unlabeled batch $\{\mathbf{u}_i\}_{i\in[B_U]}$, and EMA momentum $m$.
2: **Define:** $\mathbf{p}_i = \mathbf{p}(\mathbf{y}|\omega(\mathbf{u}_i))$
3: $\mathcal{L}_s = \frac{1}{B_L} \sum_{i=1}^{B_L} \mathcal{H}(\mathbf{y}_i, \mathbf{p}(\mathbf{y}|\omega(\mathbf{x}_i)))$       ▷ Compute $\mathcal{L}_s$ on labeled batch
4: $\hat{\mu}_b = \frac{1}{B_U} \sum_{i=1}^{B_U} \max(\mathbf{p}_i)$       ▷ Compute the mean of confidence
5: $\hat{\sigma}^2 = \frac{1}{B_U} \sum_{i=1}^{B_U} (\max(\mathbf{p}_i) - \hat{\mu}_b)^2$      ▷ Compute the variance of confidence
6: $\hat{\mu}_t = m\hat{\mu}_{t-1} + (1-m)\hat{\mu}_b$       ▷ Update EMA of mean
7: $\hat{\sigma}_t^2 = m\hat{\sigma}_{t-1}^2 + (1-m)\frac{B_U}{B_U-1}\hat{\sigma}_b^2$      ▷ Update EMA of variance
8: **for** $i = 1$ to $B_U$ **do**
9:    $\lambda(\mathbf{p}_i) = \begin{cases} \exp\left(-\frac{(\max(\mathrm{UA}(\mathbf{p}_i))-\hat{\mu}_t)^2}{2\hat{\sigma}_t^2}\right), & \text{if } \max(\mathrm{UA}(\mathbf{p}_i)) < \hat{\mu}_t, \\ 1.0, & \text{otherwise.} \end{cases}$   ▷ Compute loss weight
10: **end for**
11: $\mathcal{L}_u = \frac{1}{B_U} \sum_{i=1}^{B_U} \lambda(\mathbf{p}_i)\mathcal{H}(\hat{\mathbf{p}}_i, \mathbf{p}(\mathbf{y}|\Omega(\mathbf{u}_i)))$     ▷ Compute $\mathcal{L}_u$ on unlabeled batch
12: **Return:** $\mathcal{L}_s + \mathcal{L}_u$

---

## A.3 EXPERIMENT DETAILS

### A.3.1 CLASSIC IMAGE CLASSIFICATION

We present the detailed hyper-parameters used for the classic image classification setting in Table 6 for reproduction. We use NVIDIA V100 for training of classic image classification. The training time for CIFAR-10 and SVHN on a single GPU is around 3 days, whereas the training time for CIFAR-100 and STL-10 is around 7 days.

Table 6: Hyper-parameters of classic image classification tasks.

| Dataset | CIFAR-10 | CIFAR-100 | STL-10 | SVHN | ImageNet |
|---|---|---|---|---|---|
| Model | WRN-28-2 | WRN-28-8 | WRN-37-2 | WRN-28-2 | ResNet-50 |
| Weight Decay | 5e-4 | 1e-3 | 5e-4 | 5e-4 | 3e-4 |
| Labeled Batch size | 64 | | | | 128 |
| Unlabeled Batch size | 448 | | | | 128 |
| Learning Rate | 0.03 | | | | |
| Scheduler | $\eta = \eta_0 \cos\left(\frac{7\pi k}{16K}\right)$ | | | | |
| SGD Momentum | 0.9 | | | | |
| Model EMA Momentum | 0.999 | | | | |
| Prediction EMA Momentum | 0.999 | | | | |
| Weak Augmentation | Random Crop, Random Horizontal Flip | | | | |
| Strong Augmentation | RandAugment (Cubuk et al., 2020) | | | | |

### A.3.2 LONG-TAILED IMAGE CLASSIFICATION

The hyper-parameters for long-tailed image classification evaluation is shown in Table 7. We use Adam optimizer instead. For faster training, WRN-28-2 is used for both CIFAR-10 and CIFAR-100. NVIDIA V100 is used to train long-tailed image classfication, and the training time is around 1 day.

Table 7: Hyper-parameters of long-tailed image classification tasks.

| Dataset | CIFAR-10 | CIFAR-100 |
|---|---|---|
| Model | WRN-28-2 | |
| Weight Decay | 4e-5 | |
| Labeled Batch size | 64 | |
| Unlabeled Batch size | 128 | |
| Learning Rate | 0.002 | |
| Scheduler | $\eta = \eta_0 \cos\left(\frac{7\pi k}{16K}\right)$ | |
| Optimizer | Adam | |
| Model EMA Momentum | 0.999 | |
| Prediction EMA Momentum | 0.999 | |
| Weak Augmentation | Random Crop, Random Horizontal Flip | |
| Strong Augmentation | RandAugment (Cubuk et al., 2020) | |

### A.3.3 TEXT CLASSIFICATION

For text classification tasks, we random split a validation set from the training set of each dataset used. For IMDb and AG News, we randomly sample 1,000 data and 2,500 data per-class respectively as validation set, and other data is used as training set. For Amazon-5 and Yelp-5, we randomly sample 5,000 data and 50,000 data per-class as validation set and training set respectively. For DBpedia, the validation set and training set consist of 1,000 and 10,000 samples per-class.

The training parameters used are shown in Table 8. Note that for strong augmentation, we use back-translation similar to (Xie et al., 2020). We conduct back-translation offline before training, using EN-DE and EN-RU with models provided in fairseq (Ott et al., 2019). We use NVIDIA V100 to train all text classification models, the total training time is around 20 hours.

Table 8: Hyper-parameters of text classification tasks.

| Dataset | AG News | DBpedia | IMDb | Amazom-5 | Yelp-5 |
|---|---|---|---|---|---|
| Model | Bert-Base | | | | |
| Weight Decay | 1e-4 | | | | |
| Labeled Batch size | 16 | | | | |
| Unlabeled Batch size | 16 | | | | |
| Learning Rate | 1e-5 | | | | |
| Scheduler | $\eta = \eta_0 \cos(\frac{7\pi k}{16K})$ | | | | |
| Model EMA Momentum | 0.0 | | | | |
| Prediction EMA Momentum | 0.999 | | | | |
| Weak Augmentation | None | | | | |
| Strong Augmentation | Back-Translation (Xie et al., 2020) | | | | |

## A.4 EXTEND EXPERIMENT RESULTS

In this section, we provide detailed experiments on the implementation of the sample weighting function in unlabeled loss, as shown in Table 9. One can observe most fixed functions works surprisingly well on CIFAR-10 with 250 labels, yet Gaussian function demonstrate the best results on CIFAR-10 with 40 labels. On the SVHN with 40 labels, Linear and Quadratic function fails to learn while Laplacian and Gaussian function shows better performance. Estimating the function parameters from the confidence and making the function truncated allow the model learn more flexibly and yields better performance for both Laplacian and Gaussian function. We visualize the functions studied in Fig. 4, where one can observe the truncated Gaussian function is most reasonable by assigning diverse weights for samples whose confidence is within its standard deviation.

Table 9: Detailed results of different instantiation of $\lambda\mathbf{p}$ on CIFAR-10 with 40 and 250 labels, and SVHN-10 with 40 labels.

| Method | $\lambda(\mathbf{p})$ | Learnable | CIFAR-10 40 | CIFAR-10 250 | SVHN-10 40 |
|---|---|---|---|---|---|
| Linear | $\max(\mathbf{p})$ | - | $11.38_{\pm 3.92}$ | $5.41_{\pm 0.19}$ | $15.27_{\pm 28.92}$ |
| Quadratic | $-(\max(\mathbf{p}) - 1)^2 + 1$ | - | $12.44_{\pm 5.67}$ | $5.94_{\pm 0.22}$ | $84.11_{\pm 1.84}$ |
| Laplacian | $\exp\left(-\frac{\lvert \max(\mathbf{p}) - \mu \rvert}{b}\right), \mu = 1.0, b = 0.3$ | - | $13.29_{\pm 3.33}$ | $5.24_{\pm 0.16}$ | $12.77_{\pm 10.33}$ |
| Gaussian | $\exp\left(-\frac{(\max(\mathbf{p}) - \mu)^2}{2\sigma^2}\right), \mu = 1.0, \sigma = 0.3$ | - | $7.73_{\pm 1.44}$ | $4.98_{\pm 0.02}$ | $12.95_{\pm 8.79}$ |
| Trun. Laplacian | $\begin{cases} \exp\left(-\frac{\lvert \max(\mathbf{p}) - \mu \rvert}{b}\right), & \text{if } \max(\mathbf{p}) < \mu, \\ 1.0, & \text{otherwise.} \end{cases}$ | $\mu, b$ | $5.30_{\pm 0.09}$ | $5.14_{\pm 0.20}$ | $3.12_{\pm 0.30}$ |
| Trun. Gaussian | $\begin{cases} \exp\left(-\frac{(\max(\mathbf{p}) - \mu)^2}{2\sigma^2}\right), & \text{if } \max(\mathbf{p}) < \mu, \\ 1.0, & \text{otherwise.} \end{cases}$ | $\mu, \sigma$ | $4.91_{\pm 0.12}$ | $4.82_{\pm 0.09}$ | $2.33_{\pm 0.25}$ |

## A.5 EXTENDED ABLATION STUDY

We provide the additional ablation study of other components of SoftMatch, including the EMA momentum parameter $m$, the variance range of truncated Gaussian function, and the target distribution of Uniform Alignment (UA), on CIFAR-10 with 250 labels.

**EMA momentum**. We compare SoftMatch with momentum 0.99, 0.999, and 0.9999 and present the results in Table 10. A momentum of 0.999 shows the best results. While different momentum does not affect the final performance much, they have larger impact on convergence speed, where a smaller momentum value results in faster convergence yet lower accuracy and a larger momentum slows down the convergence.

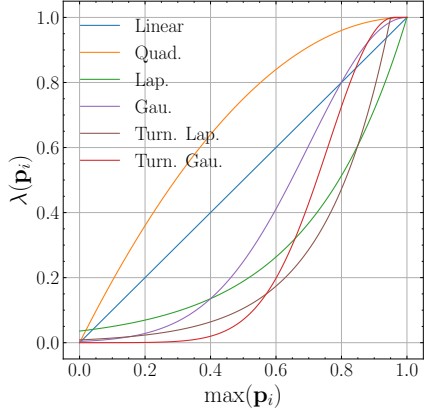

Figure 4: Sample weighting function visualization

Table 10: Ablation of EMA momentum $m$ on CIFAR-10 with 250 labels.

| Momentum | Error Rate |
|---|---|
| 0.99 | $4.92_{\pm0.11}$ |
| 0.999 | $4.82_{\pm0.09}$ |
| 0.9999 | $4.86_{\pm0.12}$ |

Table 11: Ablation of variance range in Gaussian function on CIFAR-10 with 250 labels.

| Variance Range | Error Rate |
|---|---|
| $\sigma$ | $4.97_{\pm0.13}$ |
| $2\sigma$ | $4.82_{\pm0.09}$ |
| $3\sigma$ | $4.84_{\pm0.15}$ |

Table 12: Ablation of target distribution of UA on CIFAR-10 with 250 labels.

| Target Dist. | Error Rate |
|---|---|
| $\mathbf{p}_L(\mathbf{y})$ | $4.83_{\pm0.12}$ |
| $\hat{\mathbf{p}}_L(\mathbf{y})$ | $4.90_{\pm0.23}$ |
| $\mathbf{u}(C)$ | $4.82_{\pm0.09}$ |

**Variance range**. We study the variance range of Gaussian function. In all experiments of the main paper, we use the $2\sigma$ range, i.e., divide the estimated variance $\hat{\sigma}_t$ by 4 in practice. The variance range directly affects the degree of softness of the truncated Gaussian function. We show in Table 11 that using $\sigma$ directly results in a slight performance drop, while $2\sigma$ and $3\sigma$ produces similar results.

**UA target distribution**. In the main paper, we validate the target distribution of UA on long-tailed setting. We also include the effect of the target distribution of UA on balanced setting. As shown in Table 12, using uniform distribution $\mathbf{u}(c)$ or the ground-truth marginal distribution $\mathbf{p}_L(\mathbf{y})$ produces the same results, whereas using the estimated $\hat{\mathbf{p}_L}(\mathbf{y})$ (Berthelot et al., 2021) has a performance drop.

A.6 EXTEND ANALYSIS ON TRUNCATED GAUSSIAN

In this section, we provide further visualization about the confidence distribution of pseudo-labels, and the weighting function, similar to Fig. 1(a) but on CIFAR-10. More specifically, we plot the histogram of confidence of pseudo-labels and of wrong pseudo-labels, from epoch 1 to 6. We select the first 5 epochs because the difference is more significant. Along with the histogram, we also plot the current weighting function over confidence, as a visualization how the pseudo-labels over different confidence interval are used in different methods.

Fig. 5 summarizes the visualization. Interestingly, although FixMatch adopts quite a high threshold, the quality of pseudo-labels is very low, i.e., there are more wrong pseudo-labels in each confidence interval. This reflects the important of involving more pseudo-labels into training at the beginning, as in SoftMatch, to let the model learn more balanced on each class to improve quality of pseudo-labels.

A.7 EXTEND ANALYSIS ON UNIFORM ALIGNMENT

In this section, we provide more explanation regarding the mechanism of Uniform Alignment (UA). UA is proposed to make the model learn more equally on each classes to reduce the pseudo-label imbalance/bias. To do so, we align the expected prediction probability to a uniform distribution

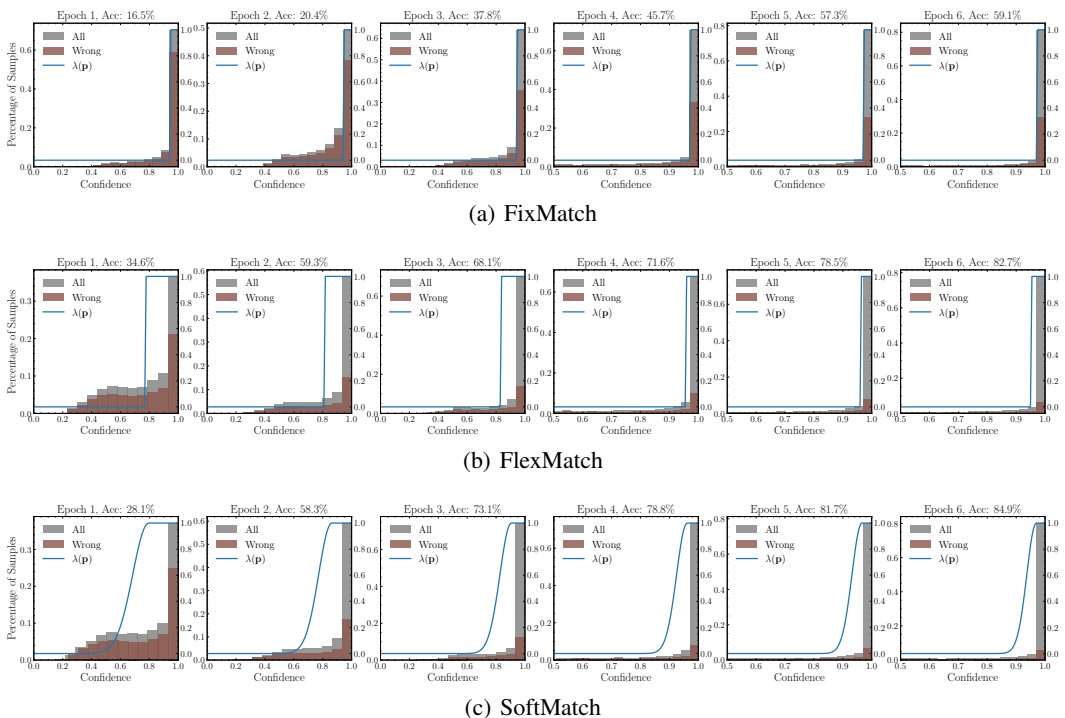

Figure 5: Histogram of confidence of pseudo-labels, learned by (a) FixMatch; (b) Flexmatch; (c) SoftMatch, for first 6 epochs on CIFAR-10. The weighting function over confidence of each method is shown as the blue curve. For FlexMatch, we plot the average threshold. SoftMatch presents better accuracy by utilizing pseudo-labels in a more efficient way.

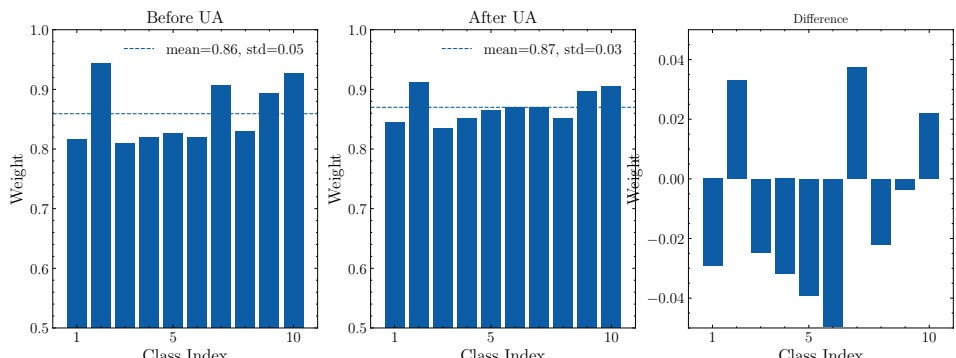

Figure 6: Average weight for each class according to pseudo-label, for (a) before UA; and (b) after UA. We also include the difference of them in (c). UA helps to balance the average weight of each class.

when computing the sample weights. A difference of UA and DA is that UA is only used in weight computing, and not used in consistency loss. To visualize this, we plot the average class weight according to pseudo-labels of SoftMatch before UA and after UA at the beginning of training, as shown in Fig. 6. UA facilitates more balanced class-wise sample weight, which would help the model learn more equally on each class.

