# OpenReview forum: "SoftMatch: Addressing the Quantity-Quality Tradeoff in Semi-supervised Learning"
_ICLR.cc/2023/Conference — ICLR 2023 poster_

### Official Review · Reviewer_QAQp · 2022-10-17

**Confidence:** 4
**Correctness:** 3
**Technical Novelty And Significance:** 3
**Empirical Novelty And Significance:** 4
**Recommendation:** 8

**Clarity, Quality, Novelty And Reproducibility:**

The paper is generally well-written. The idea is also novel. The code is provided.

**Strength And Weaknesses:**

Pros:

1. This paper is written in a clear way.

2. The idea of this work is reasonable, and the details of the proposed algorithm are provided.

3. The experiments are convincing.

I do not have major concerns on this work. I only have several questions:
1. This work targets the quantity-quality trade-off problem in semi-supervised learning, which is meaningful. I agree that in each iteration, this problem does exist. But for the entire learning process which consists of many iterations, I’m not sure whether this is still an important problem. In traditional pseudo-labeling methods, only the unlabeled examples with high confidence are utilized for each iteration. This will decrease the quantity, but the quality will be high, as most of them are with correct labels. Then, when the iterations proceed, the learning performance will be gradually improved with a large probability. However, if we spend too much efforts on the quantity-quality trade-off in each iteration, I’m afraid that this might hurt the accuracy as well as decrease the efficiency from the global view of learning. Therefore, I hope the authors can give more explanations on this point.
2. For pseudo-labeling methods, the authors are suggested to mention whether their method is transductive or inductive, or is applicable to both conditions.
3. Some typos should be fixed. For example, in the caption of Fig.1, “(b) Quality of pseudo-labels” should be “(c) Quality of pseudo-labels”.


**Summary Of The Paper:**

This paper tackles the problem of quantity-quality trade-off in pseudo-labeling based semi-supervised learning. Specifically, the authors develop a truncated Gaussian function to weight samples based on their confidence, which can be viewed as a soft version of the confidence threshold. They also enhance the utilization of weakly-learned classes by proposing a uniform alignment approach. The experiments on various types of datasets demonstrate the effectiveness of the proposed method.

**Summary Of The Review:**

I do not have major concerns on this paper. I only have several minor concerns as listed above. I hope the authors can provide the reply regarding my previous minor concerns.

---

> ### Author Response · Authors · 2022-11-12
> **Response to Reviewer QAQp**
>
> We appreciate for your positive review and we address your remaining concerns in the following.
>
> **W1: This work targets the quantity-quality trade-off problem in semi-supervised learning, which is meaningful. I agree that in each iteration, this problem does exist. But for the entire learning process which consists of many iterations, I’m not sure whether this is still an important problem. In traditional pseudo-labeling methods, only the unlabeled examples with high confidence are utilized for each iteration. This will decrease the quantity, but the quality will be high, as most of them are with correct labels. Then, when the iterations proceed, the learning performance will be gradually improved with a large probability. However, if we spend too much efforts on the quantity-quality trade-off in each iteration, I’m afraid that this might hurt the accuracy as well as decrease the efficiency from the global view of learning. Therefore, I hope the authors can give more explanations on this point.**
>
> The importance of addressing the quantity-quality trade-off lies exactly in the training process. However, it should be viewed in a dynamic perspective.
>
> At the beginning of training, the model tends to be more robust to the noise [1], thus it would be beneficial to involve more samples from unlabeled set into training (even with noise). Using a high threshold as in FixMatch will initially ensure the quality, but the very limited quantity leads to convergence issues. Besides, the model will learn more on easy samples/classes with the limited quantity, resulting in biased pseudo-labels which in return hurts the training later. This is shown in both Fig. 1(c) and Fig. 2(c), where FixMatch indeed present lower quality compared to SoftMatch.
>
> This can be visualized in Fig. 5 of Sec. A.7. in revised paper, where FixMatch indeed produces more wrong pseudo-labels and utilizes them less efficiently.
>
> As training progresses, the model learns better from the initial high quantity pseudo-labels and becomes more confident on correct samples and unconfident on possibly wrong samples. In this stage, the model should learn less on the possibly wrong samples and more on correct samples.  FixMatch/FlexMatch simply discard the low quality samples, but SoftMatch still utilize them with lower weights.
>
> To summarize, the model should focus more on quantity at the beginning of training and gradually ensures the quality along training while keeping the quantity. In SoftMatch, this is a natural and adaptive process at the training iteration with a learnable turncated Gaussian function. At the beginning of training, the model's average confidence is low, and variance is high, thus turncated Gaussian function
>
> [1] Understanding deep learning requires rethinking generalization. Chiyuan Zhang, Samy Bengio, Moritz Hardt, Benjamin Recht, Oriol Vinyals. 2016.
>
> **W2: For pseudo-labeling methods, the authors are suggested to mention whether their method is transductive or inductive, or is applicable to both conditions.**
>
> In the paper, we only consider the inductive case, where the test set is not visible to the model and not participate into training. However, it is indeed possible to apply our method (or other pseudo-labeling methods) into transductive applications, such as transductive few-shot learning.
>
> **W3: Some typos should be fixed. For example, in the caption of Fig.1, “(b) Quality of pseudo-labels” should be “(c) Quality of pseudo-labels”.**
>
> Thanks for pointing this out. We have fixed the typos we find.
>
> Hope our response could resolve your concern. Please let us know if you have further questions.

---

### Official Review · Reviewer_ZV9Z · 2022-10-23

**Confidence:** 4
**Clarity, Quality, Novelty And Reproducibility:** Generally good. But many issues shoul…
**Correctness:** 3
**Technical Novelty And Significance:** 3
**Empirical Novelty And Significance:** 3
**Recommendation:** 6

**Strength And Weaknesses:**

Pros:
* The consideration of the quantity-quality trade-off is very interesting and novel.
* The proposed solution is simple but effective. Imbalanced setting is also taken into consideration in this paper.
* The evaluation is conducted on image and text datasets, together with imbalanced settings. The performance is also promising.

Cons:
* Even with the summary of formulations in Table 1, why the proposed SoftMatch could achieve both high quality and quantity is not well-explained.
* More recent studies towards adaptive thresholding and imbalanced setting in SSL should be discussed and compared, e.g., [1-2].
* Why the truncated version of Gaussian function is used should be explained (although the ablation results show it’s the best choice).
* It would be better if the learned truncated Gaussian function and the real confidence distribution could be illustrated on real-world datasets (for several epochs).
* It is claimed that the proposed uniform alignment could alleviate the distribution of generated pseudo-labels to be imbalanced. Then [3] should be included for performance comparison. Besides, the visualization of the pseudo-label distribution (w.r.t. classes) is helpful to understand the effect of uniform alignment, especially on imbalanced datasets.
* Why only 3 random seeds are used in Table 2? In the original FixMatch paper, the number of random seeds is 5.
* It is not clear on which dataset and with which number of labeled data the ablation studies are conducted.

Ref:

[1] Class-Imbalanced Semi-Supervised Learning with Adaptive Thresholding, ICML22.

[2] Smoothed Adaptive Weighting for Imbalanced Semi-Supervised Learning: Improve Reliability Against Unknown Distribution Data, ICML22.

[3] Debiased Learning from Naturally Imbalanced Pseudo-Labels, CVPR22.


**Summary Of The Paper:**

This paper points out the inherent quantity-quality trade-off problem of pseudo-labeling with confidence thresholding exists in recent semi-supervised learning (SSL) methods. It then proposes a soft adaptive sample weighting scheme using a truncated Gaussian function to utilize both high quantity and quality of pseudo-labels. Meanwhile, a uniform alignment technique is designed for pseudo-labeling, which exploits original predictions to compute pseudo-labels and normalized predictions to compute sample weights.

**Summary Of The Review:**

The main idea is very interesting to me, and the performance is good. However, the clarity and experiments should be improved.

---

> ### Author Response · Authors · 2022-11-12
> **Response to  Reviewer ZV9Z**
>
> Thanks for your general positive feedback, recognition of the novelty and simplify of our method, and acknowledgement of the promising performance of SoftMatch. We now address your excellent suggestions to make SoftMatch more explainable and for better understanding of SoftMatch.
>
> **W1: Even with the summary of formulations in Table 1, why the proposed SoftMatch could achieve both high quality and quantity is not well-explained.**
>
> - For quantity, SoftMatch ensures a lower bound of 50\% on the quantity. Besides, the lower bound would become tighter as the model learns to be more confident.
> - For quality, although it cannot be exactly quantified, SoftMatch treat the pseudo-labels above the Gaussian mean as correct and weight the others according to their confidence. Thus in general, the quality of the pseudo-labels is also high.
> - Both high quality and quantity leads to faster convergence of SoftMatch and better performance.
>
> **W2: More recent studies towards adaptive thresholding and imbalanced setting in SSL should be discussed and compared, e.g., [1-2].**
>
> Thanks for this suggestion. We have conducted the experiments for comparing with more recent methods in imbalanced SSL.  We have also included the discussion of these imbalanced algorithms in the related work.
> The accuracy on CIFAR10-LT ($N_1=1500, M_1=3000$, $\gamma=100$) and CIFAR100-LT ($N_1=150, M_1=300$, $\gamma=10$) is shown in the table below.
>
> |  	| CIFAR10-LT 	| CIFAR100-LT 	|
> |---|---|---|
> |  $\gamma$	| 100 	| 10 	|
> | FixMatch 	| 74.89 	| 57.76 	|
> | Adsh [1]	| 77.23 	| 58.65 	|
> | SAW [2] 	| 80.93 	| 57.55 	|
> | SoftMatch 	| 77.42 	| 58.76 	|
>
> Note that while SoftMatch is *not* specifically designed for imbalanced setting, it already outperforms Adsh. SaW is better on CIFAR10-LT but inferior than SoftMatch on CIFAR100-LT.
>
> **W3: Why the truncated version of Gaussian function is used should be explained (although the ablation results show it’s the best choice).**
>
> The truncation is used mainly for ensuring the model can learn from the relatively more confident samples and ensuring the quantity of pseudo-labels to be at least 50%. Besides, it is empirically verified as the best one.
>
> **W4: It would be better if the learned truncated Gaussian function and the real confidence distribution could be illustrated on real-world datasets (for several epochs).**
>
> Thanks for this wonderful suggestion, We have added corresponding visualization on CIFAR-10 in **Fig.5** of Sec. A.7. In the visualization, SoftMatch still produce the best performance by utilizing pseudo-labels more efficiently.
>
>
> **W5: It is claimed that the proposed uniform alignment could alleviate the distribution of generated pseudo-labels to be imbalanced. Then [3] should be included for performance comparison. Besides, the visualization of the pseudo-label distribution (w.r.t. classes) is helpful to understand the effect.**
>
> Thanks for this suggestion. DebiasPL is a very recent work related to debias of the pseudo-labels, thus we add the comparison of DebiasPL and SoftMatch on CIFAR-10:
>
> |  	| CIFAR-10 40 	| CIFAR-10 250 	|
> |---|---|---|
> | DebiasPL 	| 5.23 (0.34) 	| 4.96 (0.21) 	|
> | SoftMatch 	| 4.91 (0.12) 	| 4.82 (0.09) 	|
>
>
> We will include the full comparision to DebiasPL in the next version of this paper.
>
> For UA, note that it alleviates the bias/imbalance of pseudo-labels by balancing the sample weights, instead of directly changing the pseudo-labels as in DA. Besides, we have added the visualization of pseudo-labels for better explanation of UA, as shown Fig. 6 in A. 8. The operation of UA balances the average sample weights for each class at the early stage of training SoftMatch, thus the model can equally learn on each class.
>
> **W6: Why only 3 random seeds are used in Table 2? In the original FixMatch paper, the number of random seeds is 5.**
>
> Many recent papers evaluated the SSL algorithms using 3 seeds, such as FixMatch and FlexMatch. The codebase TorchSSL we used also has a default setting of 3 seeds. This is also due to the limited computation resources. Training SSL algorithms with current settings is very time-consuming. Even CIFAR-10 needs 3 GPU days to train with V100. Other datasets only take longer.
>
> **W7: It is not clear on which dataset and with which number of labeled data the ablation studies are conducted.**
>
> Thanks for pointing this confusion out. All of out ablation (except for Fig. 3(a)) are conducted on three settings, CIFAR-10 with 40 labels, CIFAR-10 with 250 labels, and SVHN with 40 labels. Fig. 3(a) is on CIFAR-10-LT with an imbalance ratio of 50.
>
> If our response well resolves your concerns, please consider raising the score and let us know if further questions exist.

---

### Official Review · Reviewer_ZqdU · 2022-10-24

**Confidence:** 4
**Correctness:** 3
**Technical Novelty And Significance:** 3
**Empirical Novelty And Significance:** 3
**Recommendation:** 6

**Clarity, Quality, Novelty And Reproducibility:**

The paper is clearly presented and well organized. The overall quality needs improving and the novelty is not enough as there have been numerous brilliant SSL techniques. I didn't carefully check the reproducibility but I didn't doubt it.

**Strength And Weaknesses:**

Strength:
1. It is interesting that the authors understood the limitation of pseudo-labels from the view of quantity-quality trade-off, and some experimental evidence was presented by comparing with FixMatch and FlexMatch.
2. The weight function for unlabeled samples adjusted the importance of the contributions of different samples during training, which is intuitively better compared with naïve pseudo-labels.

Weaknesses:
1. The quantity-quality trade-off shown in Fig. 1(a) seems not fair. For Fixmatch, how many unlabeled samples are selected by a hard threshold is dependent on the training process. Experimentally, only a small proportion of unlabeled data will be selected at the beginning of model training, and it should select the most unlabeled data once the model converges. Otherwise, the model performance would not be guaranteed. Similar to Flexmatch. Although the motivation sounds interesting, the experimental observation is not clear or convincing to me.
2. Following 1, I felt a bit confused about why SoftMatch here is a curve instead of a point.
3. From Section 2.2, the quantity and quality functions f(p) and g(p) are only used for analysis instead of the SoftMatch method. From Table 1, I can sense the difference between them, but I cannot tell the advantages of SoftMatch directly.
4. Following 3, the comparison between Pseudo-label and FixMatch in Table 1 is unclear to me. E.g., Pseudo-Label also has the sample selection, but the weight equals a const lambda_max.
5. Eq. (2) can be viewed as a soft version of sample selection, but in what case it is needed or outperforms other strategies is not clear after I read the paper.
6. The weight is normalized at a batch level with EMA stabilizing the training, thus it shares the same insight with [1]. However, such a branch of work has not been mentioned.
7. From the experiments, the improvement of SoftMatch on standard SSL benchmarks is quite limited but it showed some potential in imbalanced class cases, which is however not the key challenge from the introduction of this work.

[1] Sinkhorn label allocation: Semi-supervised classification via annealed self-training, ICML 2021.



**Summary Of The Paper:**

This paper proposes SoftMatch to improve both the quantity and quality of pseudo-labels in semi-supervised learning. Basically, the authors designed a truncated Gaussian function to weigh the unlabeled samples and remedied the imbalanced problem by encouraging a low entropy distribution within a batch.

**Summary Of The Review:**

This work needs further improvement from both a thorough investigation of SSL and clearer demonstrations to the main contributions.

---

> ### Author Response · Authors · 2022-11-12
> **Response to Reviewer ZqdU -- Part 1**
>
> Firstly we appreciate your acknowledgment of our interpretation of pseudo-labels from the perspective of the weight function. We also recognize that your concerns are mainly the confusion and misunderstanding of Fig. 1 and Tab. 1, and we provide more explanation to address your concern in the following：
>
> **W1: The quantity-quality trade-off shown in Fig. 1(a) seems not fair.**
>
> For Fig. 1(a), we choose an iteration during training to show a better understanding of the quantity-quality trade-off. We agree with the statement that "Experimentally... Similar to FlexMatch", however this is the exact issue we are trying to solve in SoftMatch.
>
> - FixMatch's high threshold excludes most unlabeled samples at the beginning of training, and the model is more trained on the easier samples/classes with higher confidence. This will impede the performance of the model and make convergence more difficult. In return, confirmation bias will also affect the quality of pseudo-labels in later training of FixMatch. This is shown in Fig. 1(c), where FixMatch indeed presents lower quality (not high quality).
> - FlexMatch tends to solve the quantity issue in FixMatch by dynamically lowering the threshold. However, as shown in Fig. 1(a), this will inevitably involve *more wrong* predictions in training. This might not be an issue at the beginning of training but will also affect the general quality of pseudo-labels in later training, as shown in Fig. 1(c).
> - On the contrary, SoftMatch *fully* utilizes the unlabeled set, achieving the highest quantity in Fig. 1(b). It assigns lower weights to possibly incorrect samples and higher weights to correct samples, according to confidence. Thus it also achieves the highest quality in Fig. 1(c). It solves the issue of the quantity-quality trade-off present in FixMatch and FlexMatch.
>
> Besides the illustration in Fig.1, we also include a visualization on CIFAR10 in Sec. A.6, showing similar effects.
>
> **W2: Following 1, I felt a bit confused about why SoftMatch here is a curve instead of a point.**
>
> The curve in figure 1 (a) describes that: at each interval of the predicted confidence, the utilization ratio of correct pseudo-labels. Thus only SoftMatch can achieve describing the correct enrollment on every single value of confidence to make it a complete curve shown in Fig.1(a). Precisely, there would be two similar curves for FlexMatch and FixMatch which can only start from the corresponding point. Drawing in this way is to clearly illustrate that a large proportion of (correct) samples are not utilized by FlexMatch and FixMatch.
>
> **W3: From Section 2.2, the quantity and quality functions f(p) and g(p) are only used for analysis instead of the SoftMatch method. From Table 1, I can sense the difference between them, but I cannot tell the advantages of SoftMatch directly.**
>
> - The quantity-quality trade-off is a fundamental problem in SSL, and this is why their definitions are critical and indeed serve as a theoretical analysis. SoftMatch is a method addressing the quantity-quality trade-off and as we analyze in the paper as shown in Sec. 3.1, the theoretical analysis on this trade-off motivates the proposal of truncated Gaussian, which is the key component in SoftMatch.
> - The advantage of SoftMatch lies in its novel modules to address the quality-quantity trade-off: To better utilize the unlabeled data with lower confidence, we expect lower weights to be assigned to those samples. Thus the weighting function should be estimated from the predicted probability, and we choose Gaussian since it belongs to exponential family with maximum entropy property to provide a general fit to unknown distributions..
> - There are some evidence to show the advantages of SoftMatch on the Quantity-quality trade-off: From the results of SoftMatch in Tab 1., we can read about the lower bound of quantity with the usage of the truncated Gaussian function: at least **50%** quantity of pseudo-labels with an additional quantity corresponding to the estimated Gaussian are maintained.
> Note that when the estimated variance becomes smaller and predictions get more confident (naturally happens with the training going), this lower bound gets closer to 1, which indicates that the quantity utilized in SoftMatch is progressively strengthened during training.
> SoftMatch also ensures the quality of pseudo-labels. SoftMatch assigns high quality to the pseudo-labels with confidence over the Gaussian mean, and assign low weight the samples with lower confidence to constraint the quality of those samples but still utlize them. This, in general, will improve quality during training.
>
> The above analysis is not only reflected in Tab. 1, but also Fig. 1 and Fig. 2. With the proposed truncated Gaussian function for computing the weight, SoftMatch is able to achieve both high quantity and high quality when utilizing pseudo-labels and thus yields better performance.

---

> > ### Author Response · Authors · 2022-11-12
> > **Response to Reviewer ZqdU -- Part 2**
> >
> >
> > **W4: Following 3, the comparison between Pseudo-label and FixMatch in Table 1 is unclear to me. E.g., Pseudo-Label also has the sample selection, but the weight equals a const lambda_max.**
> >
> > In Table 1, the pseudo-label method represents for the very naive one without any thresholding, thus its weight equals to $\lambda_{\max}$. This is explicitly stated in Sec. 2.2 and also aligned with the implementation in FixMatch and TorchSSL.
> >
> > **W5: Eq. (2) can be viewed as a soft version of sample selection, but in what case it is needed or outperforms other strategies is not clear after I read the paper.**
> >
> > We would like to clarify some mis-understandings presented in this comment. In Eq.2, we formularize the sample weighting generally used in SSL methods as the $\lambda$ function. Eq.2 is not a newly-designed methodology but to unify the sample weighting for unlabeled data. Meanwhile, we clearly list the specific expression of $\lambda$ in Table.1 for different methods.
> > Besides, the sample selection of SoftMatch is needed nearly all the time because the issue of confirmation bias. The soft version of sample selection is a better strategies especially when the number of labels is very limited.
> >
> > We hope this addresses the reviewer's concern. To better and further discuss any concerns, we are looking forward to the reviewer's explanation on this question
> >
> > **W6: The weight is normalized at a batch level with EMA stabilizing the training, thus it shares the same insight with [1]. However, such a branch of work has not been mentioned.**
> >
> > I'm afraid there is some misunderstanding regarding this question.
> > The sample weight in SoftMatch is not normalized at batch level. After computing the sample weight from the truncated gaussian function in [0, 1], we use them directly in computing the loss. Besides, the EMA is for estimating the mean and variance parameter of the Gaussian function, which is a common solution as in BatchNorm. Thus our method does not share the same insight with [1]. But thanks for mentioning this related work, we have added discussion about it in the revised paper.
> >
> > **W7: From the experiments, the improvement of SoftMatch on standard SSL benchmarks is quite limited but it showed some potential in imbalanced class cases, which is however not the key challenge from the introduction of this work.**
> >
> > SoftMatch indeed provide some significant and promising improvement as recognized by Reviewer Fkxk, Reviewer Zv9z, and Reviewer QAQp.
> > On CV tasks, SoftMatch achieves top performance on 7 out of 11 subset evaluations, with a considerate large margin(CIFAR-100 with 400 labels by 2.84%, STL-10 with 40 labels by 7.73%, and ImageNet by 1.33%). On NLP tasks, SoftMatch also presents prominent improvement and especially for a few labeled data AGNews 40 with 2.81% performance margin compared with the second best. Besides, SoftMatch also show clear improvement on imbalanced settings, demonstrating it is a robust algorithm.
> >
> > We hope our response can resolve your concern on SoftMatch. If so, please consider raising your score. We are looking forward to your response and provide further explanations if any questions still hold.

---

> > ### Public Comment · ~Huayu_Mai1 · 2023-03-27
> > **About the upper bound of quality**
> >
> > Does the quality of pseudo-label really reach 1? It's means all the pseudo-label is correct.

---

> > > ### Author Response · Authors · 2023-03-28
> > > **Quality of pseudo-label**
> > >
> > > It depends on dataset and evaluation setting. On CIFAR-10 with 250 and 4000 labels, we observed very high quality. But on more complicated dataset such as CIFAR-100 and ImageNet, it will never be close to 1. But still, it in general provide higher quality than previous methods.

---

> ### Author Response · Authors · 2022-11-16
> **Thank you for acknowledgement of our response and score raising**
>
> Dear reviewer ZqdU,
>
> We noticed that you have raised the score from 3 to 6, which is much appreciated! Please do not hesitate to let us know if you have any other questions. Thanks!

---

### Official Review · Reviewer_Kfxk · 2022-10-24

**Confidence:** 3
**Correctness:** 3
**Technical Novelty And Significance:** 2
**Empirical Novelty And Significance:** 3
**Recommendation:** 6

**Clarity, Quality, Novelty And Reproducibility:**

This paper is well-presented and organized. The proposed idea is simple but effective. The authors provide an excellent code for experiment result reproduction.

**Strength And Weaknesses:**

Strengths:

1. This paper demonstrates the importance of the unified weighting function by formally defining the quantity and quality of pseudo-labels, and the trade-off between them.
2. This paper proposes SoftMatch to effectively leverage the unconfident yet correct pseudo-labels, fitting a truncated Gaussian function in the distribution of confidence, which overcomes the trade-off.
3. This paper demonstrates that SoftMatch outperforms previous methods on various image and text evaluation settings.


Weaknesses:

1. One issue of the existing SOTA SSL methods is the issue of efficiency (e.g., Fixmatch is time-consuming for training). Based on figure 2, the proposed SoftMatch has a similar issue.
2. What is the motivation for Uniform Alignment (UA)? Why the model prediction is aligned to a uniform distribution?


**Summary Of The Paper:**

This paper focuses on the traditional Semi-Supervised Learning (SSL) problem. The authors first revisit the popular pseudo-labeling methods via a unified sample weighting formulation and demonstrate the inherent quantity-quality trade-off problem of pseudo-labeling with thresholding, which may prohibit learning. To this end, The authors propose SoftMatch to overcome the trade-off by maintaining both high quantity and high quality of pseudo-labels during training, effectively exploiting the unlabeled data. The authors derive a truncated Gaussian function to weight samples based on their confidence, which can be viewed as a soft version of the confidence threshold. The authors further enhance the utilization of weakly-learned classes by proposing a uniform alignment approach.



**Summary Of The Review:**

See *Strength And Weaknesses*

---

> ### Author Response · Authors · 2022-11-12
> **Response to Reviewer Kfxk**
>
> Thanks for the valuable review and acknowledgement for the importance of the topic we addressed. Regarding your concern about the algorithm efficiency and the motivation of UA, our answers are the following.
>
> **W1: Issue of efficiency (e.g., Fixmatch is time-consuming for training).**
>
> The issue of efficiency of SOTA SSL methods mainly lies on two folds: (1) slow convergence of consistency regularization loss; and (2) very long training epochs. Consistency is known to be hard to converge. As pointed out in [1], there are many possible labeling solutions for unlabeled data, using consistency regularization leads to different sub-optimal optimization results rather than finding an optimal one. Besides, the training iterations of FixMatch is set to 1000 epochs with 1024 iterations per epoch, which is a long training setting.
>
> Although we use the same training setting as FixMatch for fair comparison, SoftMatch is able to facilitate the consistency-based unlabeled loss to converge faster with higher utilization rate of the pseudo labels but lower pseudo label errors, as demonstrated in Fig. 4 in main paper. To demonstrate more clearly the benefit of SoftMatch, we compare the best accuracy, and the iteration that best accuracy is achieved, and the accuracy at 20k-th iteration on CIFAR-100 with 400 labels (seed 0):
>
>
> | Method    	| Best Acc. 	| Best Iter. 	| 20k-th Acc 	|
> |-----------|-----------|------------|------------|
> | UDA       	| 54.63     	| 56k        	| 49.31      	|
> | FixMatch  	| 53.01     	| 78k        	| 45.77      	|
> | FlexMatch 	| 57.98     	| 65k        	| 54.63      	|
> | SoftMatch 	| **61.93** 	| 61k        	| **56.67**  	|
>
>
> As shown in the table, SoftMatch achieves the **best accuracy** with the **least training iterations**. Furthermore, at 20k-th iteration of training, SoftMatch also presents the superior performance, compared to previous methods.
> Thus SoftMatch is more efficient than FixMatch and previous SOTA SSL methods.
>
> [1] There Are Many Consistent Explanations of Unlabeled Data: Why You Should Average. Ben Athiwaratkun, Marc Finzi, Pavel Izmailov, Andrew Gordon Wilson. 2019.
>
>
> **W2: Motivation for Uniform Alignment (UA)**
>
> Uniform Alignment is used to alleviate the issue of *pseudo-label bias*. Since the model tends to learn some easy classes better than hard/difficult classes, it tends to generate high confidence on these easy classes and low confidence on hard classes. Thus the resulting pseudo-labels can be very class-imbalanced (easy classes have more pseudo-labels because of the general high confidence.) If we directly use these pseudo-labels, the more difficult classes will be less learned with lower weights. UA can solve this issue by suppressing the weights of pseudo-labels belonging to easy classes and enlarging the weights of difficult classes to let them participate more in training. We align the averaged predicted probability to uniform since we expect each class should be equally learned to achieve better generalization performance.
> The visualization of the effect of UA is shown in Fig.6 of A.7.
>
> If the above response resolved your concerns, please consider raising your score:) Please let us know if you still have further questions.

---

### Decision · Program_Chairs · 2023-01-20

**Decision:**

Accept: poster

**Justification For Why Not Higher Score:**

The paper presents a novel SSL method that is primarily justified though thorough empirical investigation and ablation studies. The authors clearly have a strong intuition around the problem, which if they could distill into a more rigorous theoretical analysis would boost the score of this paper further.

**Justification For Why Not Lower Score:**

The thorough empirical analysis provides enough evidence around the usefulness of the proposed algorithm and warrants publication.

**Metareview: Summary, Strengths And Weaknesses:**

This work proposes a semi-supervised learning (SSL) algorithm which introduces a per-example weight, which can be thought of as a soft-version of thresholding (i.e. filtering) examples that are low quality. In this way, the algorithm balances quantity (not throwing away points) and quality (not over-fitting to poorly pseudo-labeled examples).

The authors provide a thorough empirical analysis of the proposed method on image and text classification tasks and compare 8 different existing SSL methods. In addition to the presented accuracy improvements, the authors also provide a deeper analysis in terms of the amount of pseudo-labeled data used and the quality of that data compared to other methods as well as an ablation study to help justify certain design choices (e.g. truncated Gaussian weighting function).

All reviewers lean towards accepting the paper, with several comments (e.g. running time, clarity of some results, additional baselines) addressed during the discussion. I recommend the paper to be accepted and ask the authors to incorporate all the clarifications made during the discussion phase into the final paper.

**Note From Pc:**

if the above contains the word "oral" or "spotlight" please see: "oral" presentation means -> notable-top-5% and "spotlight" means -> notable-top-25%. As stated in our emails, we are disassociating presentation type from AC recommendations